# Modular cell-internalizing aptamer nanostructure enables targeted delivery of large functional RNAs in cancer cell lines

David Porciani [1,2,3], Leah N. Cardwell[1], Kwaku D. Tawiah[2,3], Khalid K. Alam[2,3,4], Margaret J. Lange[1,2], Mark A. Daniels[1] & Donald H. Burke [1,2,3,5]

Large RNAs and ribonucleoprotein complexes have powerful therapeutic potential, but effective cell-targeted delivery tools are limited. Aptamers that internalize into target cells can deliver siRNAs (<15 kDa, 19–21 nt/strand). We demonstrate a modular nanostructure for cellular delivery of large, functional RNA payloads (50–80 kDa, 175–250 nt) by aptamers that recognize multiple human B cell cancer lines and transferrin receptor-expressing cells. Fluorogenic RNA reporter payloads enable accelerated testing of platform designs and rapid evaluation of assembly and internalization. Modularity is demonstrated by swapping in different targeting and payload aptamers. Both modules internalize into leukemic B cell lines and remained colocalized within endosomes. Fluorescence from internalized RNA persists for ≥2 h, suggesting a sizable window for aptamer payloads to exert influence upon targeted cells. This demonstration of aptamer-mediated, cell-internalizing delivery of large RNAs with retention of functional structure raises the possibility of manipulating endosomes and cells by delivering large aptamers and regulatory RNAs.

---

[1] Department of Molecular Microbiology & Immunology, University of Missouri, Columbia, MO 65212, USA. [2] Bond Life Sciences Center, University of Missouri, Columbia, MO 65211, USA. [3] Department of Biochemistry, University of Missouri, Columbia, MO 65211, USA. [4] Department of Chemical & Biological Engineering, Northwestern University, Evanston, IL 60208, USA. [5] Department of Bioengineering, University of Missouri, Columbia, MO 65211, USA. Correspondence and requests for materials should be addressed to D.P. (email: porcianid@missouri.edu) or to D.H.B. (email: burkedh@missouri.edu)

Aptamers are increasingly investigated as diagnostic and therapeutic tools due to their ability to recognize a variety of molecular targets with high affinity and specificity[1]. These nucleic acids can serve as activating ligands[2,3], as antagonists[4,5], or as vehicles to deliver drugs and imaging agents[6,7]. Aptamers that bind cell surface markers that are preferentially expressed on specific cells are known as cell-targeting aptamers[8–10]. The subset of cell-targeting aptamers that internalize via receptor-mediated endocytosis are often termed cell-internalizing aptamers[8]. These aptamers have high potential for delivery of therapeutic payloads, including RNAs and ribonucleoprotein (RNP) complexes.

Several classes of RNAs and RNPs have shown great potential as novel therapeutic agents, including small interfering RNAs (siRNAs), microRNAs (miRNAs), antisense oligonucleotides (ASOs), aptamers, messenger RNAs (mRNAs), long non-coding RNAs (lncRNAs), and CRISPR guide RNAs (gRNAs) co-delivered with Cas9[11]. Several of these can potentially act against genes and gene products that are not currently druggable by taking advantage of high selectivity for intracellular targets. Many effective formulations have been used to deliver small RNAs (20–40 nt) with high specificity[1,12]. However, with the advent of CRISPR/cas9 and the growing interest in aptamers and other RNAs to modulate biological processes, new approaches have emerged to develop tools to deliver even larger RNAs (>100 nt) or RNP complexes[11]. Cell-internalizing aptamers have been used for targeted delivery of small molecules such as chemotherapeutic drugs[6] (<1 kDa), short therapeutic oligos (siRNAs, miRNAs, and ASOs)[13–15] (<15 kDa), and relatively large non-oligonucleotide payloads, such as toxins[16,17] (~30 kDa). However, aptamer-mediated targeted delivery of larger functional RNAs into endosomes or cytosols of diseased cells has not yet been reported. A critical consideration for this strategy is that the structured nucleic acid modules retain proper folding within the delivery platform. The cell-internalizing aptamer should preserve its cell-targeting and uptake properties without interference from the payload RNA. Reciprocally, to the extent that cellular function of the payload RNA derives from its folded 3D structure, it should retain that structure to exhibit its effects in the endosome, cytosol, or nucleus, without interference from the targeting aptamer. We show here that fluorogenic RNA aptamers can be used as surrogates for other large RNA payloads with comparable size to accelerate screening of nanostructure designs and to monitor retention of folding and function of both cell-targeting and payload aptamers. The benefits of this experimental platform are two-fold: the light-up properties of these RNA payloads are sensitive to structural variations and reveal potential RNA degradation or perturbations in aptamer folding within the nanostructure, while their successful delivery into targeted cells can be readily detected by flow cytometry and fluorescence microscopy.

The Spinach and Mango families of fluorogenic RNA aptamers are especially promising for live cell applications[18–20]. Aptamers in the Spinach family fold around a G-quadruplex[21,22] and bind a small, cell-permeable molecule that is structurally similar to the green fluorescent protein (GFP) chromophore. This molecule is

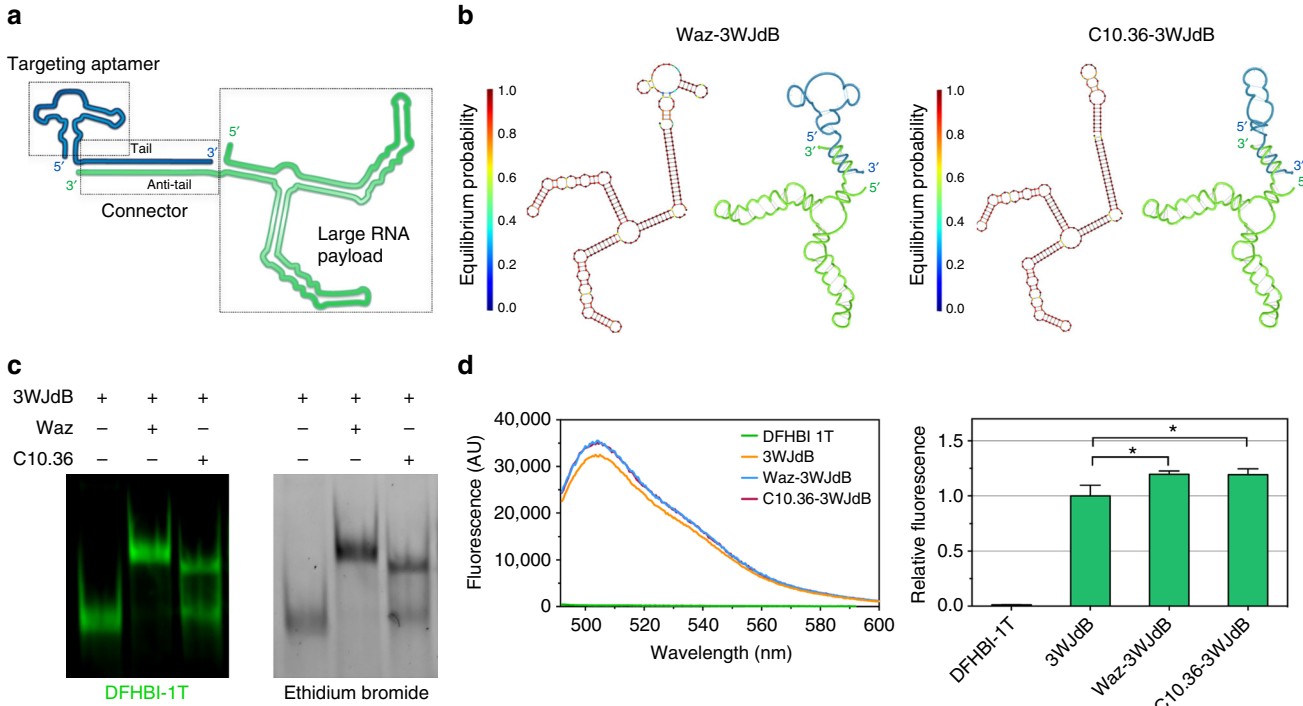

**Fig. 1** Design and assembly of cell-targeting and light-up aptamers. **a** Schematic representation of the aptamer nanostructure highlights two aptamer modules (targeting and payload) annealed via a connector domain. **b** Secondary structure predictions of Waz–3WJdB and C10.36–3WJdB. For each aptamer–aptamer hybrid, the predicted hybridization of the two modules based on NUPACK calculations is reported on the left, and a helical depiction is shown on the right. **c** Non-denaturing gel electrophoresis and dual staining with DFHBI-1T and ethidium bromide show effective formation of Waz–3WJdB and C10.36–3WJdB, and retention of the light-up properties of 3WJdB upon hybridization. Uncropped gel is shown in Supplementary Fig. 1. **d** Fluorescence spectroscopy in solution of free 3WJdB, Waz–3WJdB, and C10.36–3WJdB ($\lambda_{ex} = 472$ nm; $\lambda_{em} = 492$–600 nm). Fluorescence emission of 3WJdB (0.5 µM) was measured upon refolding in a buffer supplemented with DFHBI-1T (20 µM) either in the absence or presence of 3-fold molar excess of cell-targeting aptamers relative to 3WJdB. All curves are represented as averages of five independent experiments. Note that blue and magenta curves are overlapping. To calculate the relative fluorescence, we first integrated the fluorescence area from 495 to 600 nm of each sample, and then normalized for the fluorescence of free 3WJdB over this range. Significance was analyzed by one-way ANOVA with post hoc Tukey's test (*$p < 0.05$)

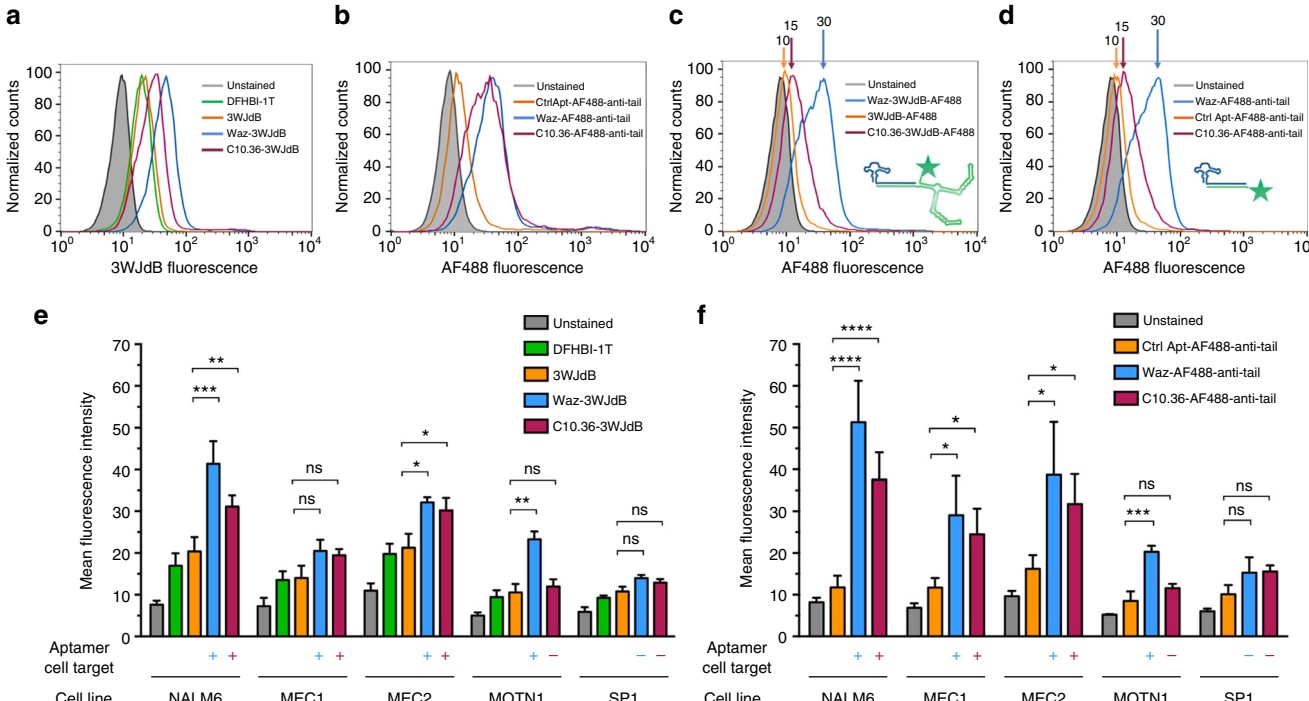

**Fig. 2** Aptamer-mediated targeted delivery of large and small RNAs. Cell-targeting aptamers, Waz and C10.36, and their respective controls were assembled with 3WJdB (**a**) or AF488-anti-tail (**b**), and their ability to bind the same batch of NALM6 cells was assessed after 1 h incubation by flow cytometry. Similarly, Waz and C10.36, and their respective controls were annealed with AF488-labeled 3WJdB (**c**) or AF488-anti-tail (**d**), and their targeting properties were assessed using the same batch of NALM6 cells. Representative flow cytometry curves illustrate a shift in fluorescence for aptamer nanostructures bearing both Waz (blue) and C10.36 (magenta) assembled with the RNA payloads. Cells treated only with DFHBI-1T (20 μM) are shown in green. All non-targeted controls are shown in orange: free 3WJdB in **a**, control aptamer assembled with AF488-anti-tail in **b,d**, free 3WJdB-AF488 in **c**. Gray filled curves: unstained cells. Normalized cell counts are reported on the y-axis, while on the x-axis is shown a log scale of fluorescence intensity. Geometric mean fluorescence intensity of Waz (blue), C10.36 (magenta), and non-targeted controls (orange) are shown above the respective curves in **c**, **d**. All curves in **c**, **d** are representative of two independent experiments. Geometric mean fluorescence intensity values for leukemia cell lines are reported in **e** using 3WJdB stained with DFHBI-1T as payload and in **f** using AF488-anti-tail as payload. Representative flow cytometry curves for MEC1, MEC2, MOTN1, and SP1 cell lines are shown in Supplementary Fig. 6. Values are the mean ± SD for at least three independent experiments. Statistical analysis for comparing multiple groups in each cell line was analyzed by one-way ANOVA with post hoc Tukey's test. Brackets with asterisks represent statistical difference: ns not significant; $*p < 0.05$; $**p < 0.01$; $***p < 0.001$; $****p < 0.0001$

poorly fluorescent in solution but becomes highly fluorescent upon the formation of a complex with the aptamer[18]. Several enhanced variations of the Spinach aptamer, such as "Broccoli", have been recently generated[19,23,24], along with the introduction of an improved GFP-like fluorophore, (Z)-4-(3,5-difluoro-4-hydroxybenzylidene)-2-methyl-1-(2,2,2-trifluoroethyl)-1H-imidazol-5(4H)-one (DFHBI-1T)[25]. Variations of these aptamers have been used as fluorescent reporters of native RNA trafficking[26], output for engineered genetic circuits[27–29], tools to monitor RNA transcription[30,31], and fluorescent sensors for metabolites[32,33]. However, only a few reports have described the use of these or other fluorogenic RNAs (e.g., Malachite green aptamer)[34] as sensors to assess preservation of their original folding within RNA nanoparticles in a cell-free context[35–38] or to monitor RNA degradation in live cells[39].

Here we show results from the design and characterization of a modular nanostructure for the aptamer-mediated intracellular delivery of large functional RNA payloads (~50–80 kDa) in live cells. This nanostructure displays a targeting aptamer module and a payload aptamer module, and the two modules self-assemble via a double-stranded connector sequence similarly to previous aptamer–siRNA or bispecific aptamer hybrids[13,40–42]. Designs are evaluated using two different targeting moieties. The Waz aptamer[43] is a 2′F-pyrimidine-modified RNA that binds human transferrin receptor (hTfR) on rapidly proliferating cells,

including most cancer cells, while the C10.36 aptamer[44] is a compact, G-quadruplex DNA that internalizes into B cell cancer cell lines upon binding an as-yet unknown cell surface molecule. Our approach exploits reporter RNA payloads, including an improved dimeric Broccoli (dB) aptamer (176 nt, in the form used here) that we recently engineered[29] and a novel trimeric version (244 nt). Unlike small molecule dyes, fluorescence signal in this system requires folded, fully-functional Broccoli aptamer modules. As such, signal from the fluorogenic aptamer directly reports on the integrity, structural stability, and intracellular processing of the RNA payload. This work highlights the application of fluorogenic RNA aptamers as real-time reporters to assess retention of correct aptamer folding within the nanostructure both after assembly and upon endocytosis into cancer cell lines, thus verifying the effective aptamer-mediated targeted delivery of much larger functional RNAs than has previously been reported.

## Results

**Design and assembly of a cell-internalizing aptamer platform.** We designed oligonucleotide nanostructures to facilitate assembly of a cell-internalizing aptamer (targeting module) with a large functional RNA (payload module) and to enable modular, targeted delivery by using appropriate aptamers to direct cell

specificity of the RNA payload. To achieve these goals, an anti-hTfR RNA aptamer (containing 2′F pyrimidines), named Waz[43], and a DNA aptamer, named C10.36[44], that specifically binds B cell cancer cell lines, were chosen for their well-studied targeting properties, sequence/structure/activity relationships, and fast intracellular uptake in specific cancer cells[43,44]. A dB aptamer variant was used as reporter RNA payload, embedded within two arms of the thermodynamically stable three-way junction (3WJ) architecture of the phi29-pRNA to yield a functional RNA aptamer, named 3WJdB[29]. Filonov et al.[45] previously reported "F30−2xBroccoli" that contains two Broccoli aptamers incorporated in arms 1 and 2 of the same 3WJ RNA scaffold. In our design, the two monomers are within arms 1 and 3, which are oriented at approximately 180° from each other, to reduce chromophore–chromophore and inter-helical RNA interactions between the two Broccoli monomers[29].

Nanostructures were formed by annealing the targeting and payload aptamers via 3′ extensions on each, denoted as "tail" on the targeting aptamer and "anti-tail" on the payload RNA (Fig. 1a). Design parameters were sought to minimize misfolding of the two modules and were guided by RNA secondary structure predictions of each module using the web-based NUPACK software suite[46] (Fig. 1b). Targeting aptamer was in excess over the payload (3:1 molar ratio) to ensure that the observed fluorescence emerged primarily from annealed aptamer. Assembly was evaluated using a native gel that was stained first with DFHBI-1T and imaged to identify bands corresponding to functional, fluorescent 3WJdB RNA. The same gel was subsequently stained with ethidium bromide to locate all molecular species (Fig. 1c). Bands with reduced electrophoretic mobility appeared upon incubation of 3WJdB with either Waz or C10.36, indicating the formation of aptamer–aptamer nanostructures in which 3WJdB retained folding and fluorescence (Fig. 1c). Waz–3WJdB complexes annealed with slightly higher efficiency than C10.36–3WJdB complexes (Fig. 1c and Supplementary Fig. 1). 3WJdB consistently annealed more efficiently with 2′F-Py RNA delivery modules than with DNA modules, even though all of these sequences carried identical 3′ tail sequences (Supplementary Fig. 1), consistent with relative stabilities of RNA/RNA and RNA/DNA duplexes[47]. These results suggest that hybridization thermodynamics of the connector domain govern annealing of the two modules.

Fluorescence of 3WJdB before and after annealing was measured in solution by fluorescence spectroscopy. In the presence of an equimolar amount of 3WJdB per sample, fluorescence of free 3WJdB was equivalent to that observed upon assembly with either Waz or C10.36 (Fig. 1d), indicating minimal impact of the cell-targeting aptamers on productive 3WJdB folding. The slight, albeit significant, increase of 3WJdB fluorescence intensity (~1.2-fold) upon annealing (Fig. 1d) suggests overall stabilization by the double-helical connector domain. Together these results indicate that this aptamer-based nanostructure assembles efficiently, that it is modular and interchangeable with respect to individual components, and that the light-up properties of 3WJdB reveal retention of productive RNA fold upon assembly.

**Cell-targeting aptamers deliver small and large RNA payloads**. Aptamers Waz and C10.36 were evaluated using flow cytometry for their abilities to deliver large RNA payloads to specific cancer cell lines. Because aptamer C10.36 binds human B cell cancer cell lines via an unidentified surface antigen, its targeting properties were assessed using a B cell acute lymphoblastic leukemia cell line, NALM6. Conveniently, this cell line also overexpresses hTfR, the receptor recognized by Waz, making it suitable for evaluating

both aptamers. Payload-targeting aptamer complexes (0.5 μM) were assembled as above and incubated with target or control cells. Stringent conditions (detailed in Methods) were used to minimize potential receptor-independent uptake via macropinocytosis and other mechanisms[48], which can be driven by long incubation times and high concentration of aptamer–aptamer complexes.

When NALM6 cells were treated with C10.36–3WJdB or Waz–3WJdB complexes and stained with DFHBI-1T, mean fluorescence intensities (MFIs) were significantly increased relative to unstained cells (~4-fold and ~5-fold) or DFHBI-1T only-treated cells (~2-fold and ~2.5-fold) (Fig. 2a, e). Importantly, when the targeting module was omitted and cells were treated with 3WJdB+DFHBI-1T only, MFIs were equivalent to those observed for DFHBI-1T-stained cells, indicating little or no cell binding by functional 3WJdB alone.

AF488-labeled RNA anti-tail (21 nt, denoted AF488-anti-tail) served as a "small-payload" control. MFIs for NALM6 cells stained with C10.36–AF488-anti-tail or Waz–AF488-anti-tail were ~3.5-fold and ~4.5-fold higher, respectively, than cells stained using non-binding control aptamer (ctrl Apt) annealed to the same payload (Fig. 2b, f). Cell staining by this control sequence was ~1.5-fold over that of unstained cells, indicating minor non-specific binding of the small RNA that was not evident for the large RNA payload.

**Aptamer-mediated delivery is not affected by RNA payload size**. Flow cytometry analysis was performed to evaluate the relative impacts of large (3WJdB) and small (AF488-anti-tail) RNA payloads on cell recognition by the C10.36 and Waz aptamers. The two fluorescent reporters above (i.e., 3WJdB–DFHBI-1T and AF488) exhibited markedly different signal strengths when analyzed by fluorescence plate reader using standard flow cytometry settings due to differences in their optimal absorbance and emission wavelengths (see Supplementary Fig. 2, Supporting text, and refs. [25,49]). To minimize the difference in intrinsic fluorescence intensities between the large and small RNA payloads, aptamer 3WJdB was covalently labeled with AF488 for this analysis. Cells were incubated with aptamer complexes as above, but without further addition of DFHBI-1T. When identical amounts of each AF488-labeled RNA were used as input, both RNA payloads stained NALM6 cells to a similar magnitude, irrespective of payload size. This size independence was observed upon delivery by either C10.36 or Waz (Fig. 2c, d). To explore delivery of even larger RNA payloads, a trimeric Broccoli aptamer was generated within the 3WJ-anti-tail architecture (3WJtriB, 244 nt) (Supplementary Fig. 3). 3WJtriB fluorescence was retained within aptamer nanostructures upon annealing to either Waz or C10.36, with a slight increase of signal relative to free 3WJdB (Supplementary Fig. 4). When NALM6 cells were treated with Waz–3WJtriB or C10.36–3WJtriB, both nanostructures efficiently delivered the larger RNA payload. Similar MFIs were measured with either cell-targeting aptamer for 3WJdB and 3WJtriB aptamer–aptamer nanostructures (Supplementary Fig. 5). Overall, these results demonstrate that C10.36 and Waz can efficiently direct delivery of large RNA payloads to the NALM6 B cell leukemia cell line, and that the size of the RNA payload up to at least 244 nt does not affect the cell-targeting properties within our aptamer nanostructure.

**Targeting large RNA payload to specific leukemia cell types**. We next evaluated the specificity of aptamer-mediated large RNA delivery with respect to cancer type, tissue-of-origin, and species. Specific cell labeling was observed for both MEC-1 and MEC-2 cells (human B cell leukemia cell lines) upon incubation with

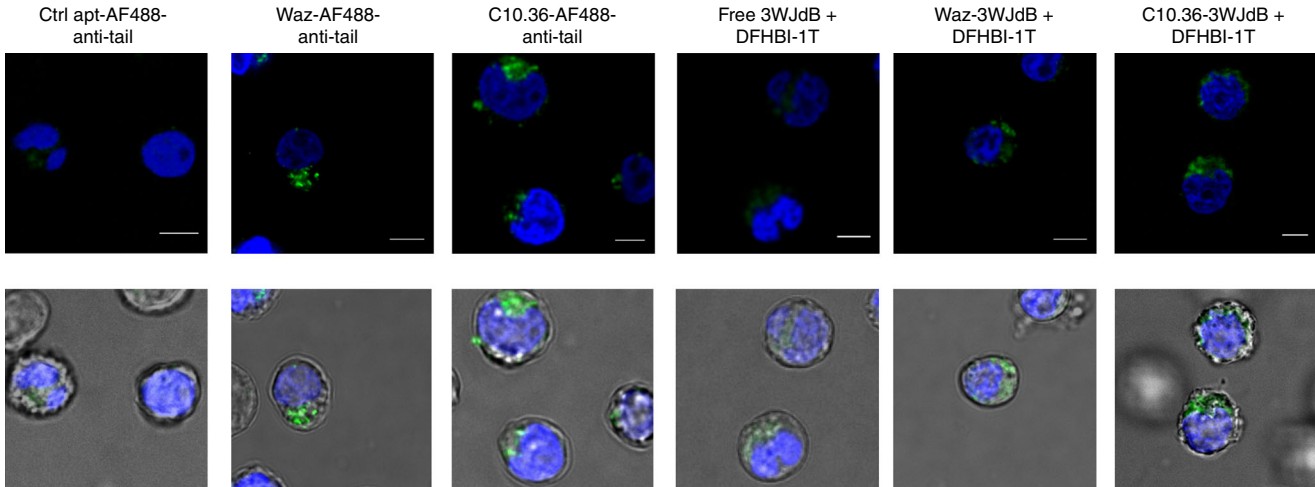

**Fig. 3** Aptamer-mediated endocytosis of small and large RNA payloads. Cell-targeting aptamers, Waz and C10.36, and their respective controls were assembled with either 3WJdB or AF488-anti-tail (green), and their internalization properties were assessed after 1 h incubation with NALM6 cells. Representative confocal microscopy images of fixed NALM6 cells show perinuclear vesicular signals (green) relative to both AF488-anti-tail and 3WJdB (0.5 μM) upon assembly with 3-fold molar excess of Waz and C10.36. In contrast, cells incubated with the non-targeted controls (control aptamer loaded with AF488-anti-tail or free 3WJdB) show minimal intracellular fluorescence. Nuclei of cells were stained with 10 μg/ml of Hoechst 33342. Merge of blue and green fluorescence is shown on the top, while the overlay between fluorescence and bright field is shown on the bottom. Cells incubated with 3WJdB samples were kept in cell-binding buffer supplemented with DFHBI-1T (20 μM) both during the 1 h incubation and the confocal microscopy imaging. Images are representative of three independent experiments. Scale bars: 5 μm. For each sample, images displaying larger field of view are shown in Supplementary Fig. 12

C10.36–AF488-anti-tail and Waz–AF488-anti-tail (Fig. 2e, f and Supplementary Fig. 6), and to a lesser extent upon incubation with C10.36–3WJdB and Waz–3WJdB (Fig. 2e, f and Supplementary Fig. 6). In contrast, MOTN-1 cells (hTfR-expressing cell line derived from human T cell leukemia) were efficiently labeled by both payloads when delivered by Waz, but not when delivered by C10.36 (Fig. 2e, f and Supplementary Fig. 6). This result establishes that C10.36 specifically binds B cell leukemia and lymphoma, but not malignant T cells. Similarly, both payloads selectively labeled HeLa cells (hTfR-expressing human cervical cancer cell line) when delivered by Waz, but not when delivered by C10.36 (Supplementary Fig. 7). Aptamer selectivity was further observed for canine lymphoma B cell line, CLL17 (labeling with AF488-anti-tail above background when delivered by C10.36 but not when delivered by Waz), while neither C10.36 nor Waz specifically directed AF488-anti-tail to the canine leukemia T cell line, CLGL-90 (Supplementary Fig. 8). No significant staining of control SP1 cells (mouse B cell leukemia cell line) was observed with payloads for Waz or C10.36 nanostructures, compared to the corresponding controls (Fig. 2e,f and Supplementary Fig. 6), consistent with reported species specificities of these aptamers toward human but not mouse cells[43,44]. These results establish that Waz and C10.36 retain their original folding and bind several target cancer cells via their corresponding cell-surface markers, even when assembled with a large RNA payload.

**3WJdB signal correlates with target expression level.** While hTfR is expressed on most human cells, its expression level can vary among cell lines and be further modulated in cell culture by cell density and other factors. To evaluate quantitative relationships between large RNA delivery and surface antigen expression, we measured hTfR levels by flow cytometry using allophycocyanin (APC)-labeled anti-CD71 antibody (anti-CD71-APC). Waz-dependent cell-labeling positively correlated ($R^2 \geq 0.90$ and Pearson's correlation coefficient $r \geq 0.95$) with hTfR expression in all four human B cell and T cell leukemia cell lines tested for both

small and large RNA payloads (Supplementary Fig. 9). Similarly, culturing NALM6 cells to high density, which is known to reduce cell proliferation and TfR expression[50,51], yielded ~2.5-fold reduction of hTfR levels measured with anti-CD71 antibody (MFI ~500 for high hTfR and ~200 for low hTfR) and ~3-fold reduction of MFI values for both Waz-488 and Waz–3WJdB (Supplementary Fig. 10). In contrast, only minimal variations were observed for background staining when using either non-binding control aptamer or free 3WJdB, irrespective of hTfR levels (Supplementary Figs. 9c and 10). Similar modulation of the cellular target of aptamer C10.36 was not possible, as this target is not yet known. Nevertheless, it is highly probable that the observed variations of C10.36-dependent cell binding for a given set of cells (Fig. 2 and Supplementary Fig. 6) may reflect differences in the expression of this unknown target. Moreover, we have noticed reduced C10.36-dependent cell staining at higher cell passages using the same batch of growing NALM6 cells, again suggesting changes in the expression levels of this unknown target over time.

**Aptamer-mediated internalization of large RNA payload.** We next used 3WJdB in fixed-cell confocal microscopy to determine whether the cell labeling described above is due to probe internalization or to surface capture only. We first determined that light-up properties of 3WJdB were not affected by paraformaldehyde (Supplementary Fig. 11). Fixed NALM6 cells were then imaged after 1 h incubation with aptamer samples (Fig. 3 and Supplementary Fig. 12). Both targeting aptamers yielded similar perinuclear punctate patterns when annealed to either the small or large RNA payload (Fig. 3), consistent with their accumulation within endosomal trafficking vesicles. Negligible signal was observed in cells stained with non-binding control aptamer loaded with AF488-anti-tail (Fig. 3). The weak background fluorescence that is observed upon treatment with DFHBI-1T (with or without free 3WJdB) (Fig. 3 and Supplementary Fig. 12) does not follow a punctate distribution, but appears instead to be more diffuse throughout the cytosol, in

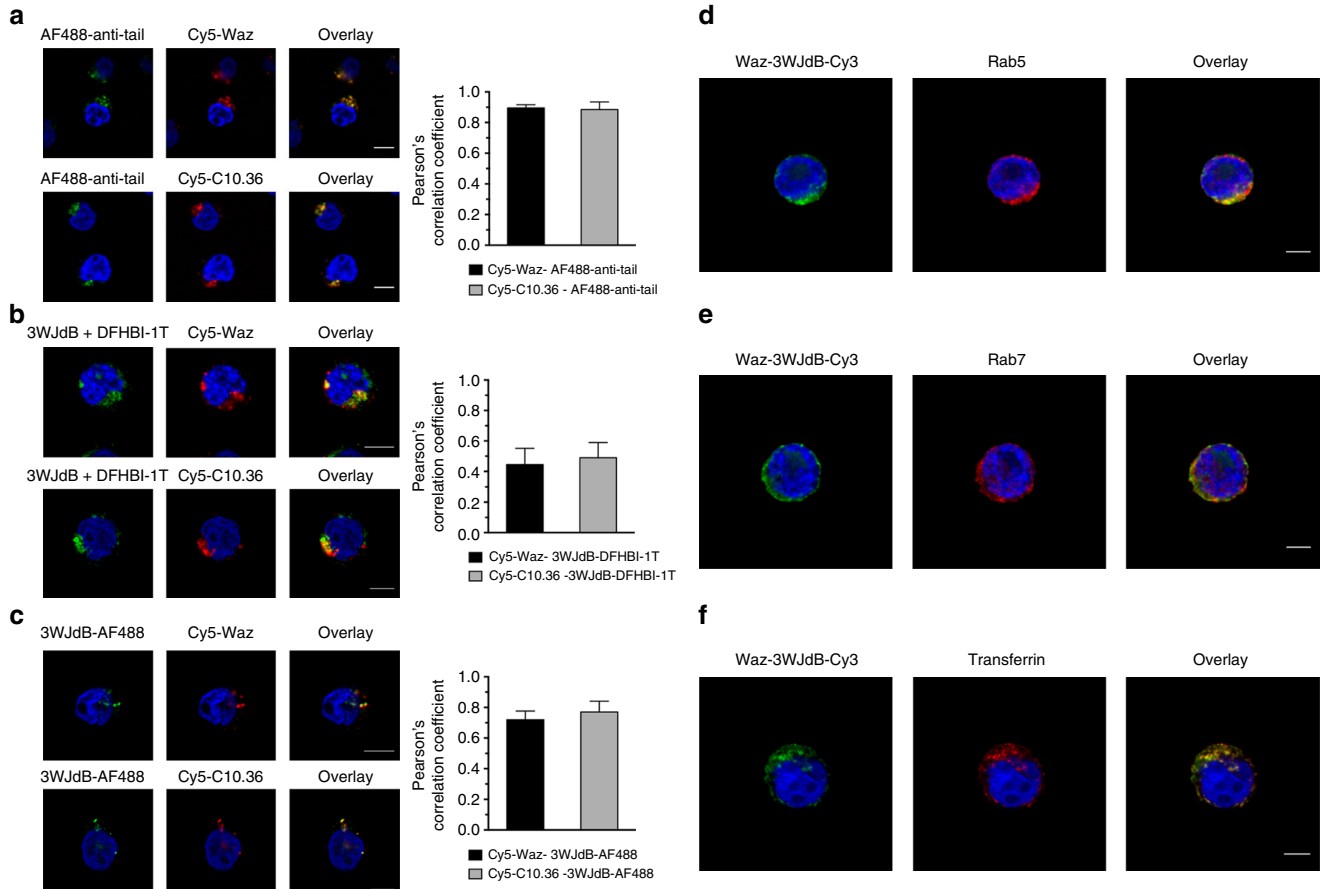

**Fig. 4** Targeting and payload aptamers colocalize within endosomes. **a**–**c** AF488-anti-tail or 3WJdB (0.5 μM, green) were assembled with 3-fold molar excess of Cy5-labeled Waz and C10.36 (red), and colocalization of the two aptamer modules was assessed after 1 h-incubation in NALM6 cells. **a** Representative confocal microscopy images of fixed NALM6 cells show significant colocalization between targeting aptamers and AF488-anti-tail. **b** A reduction of colocalization between targeting and payload modules was found using 3WJdB as a consequence of reduced brightness and photostability of 3WJdB–DFHBI-1T compared with AF488 and Cy5, as well as a higher fluorescence background due to the unbound DFHBI-1T. **c** Strong colocalization between 3WJdB and either Waz or C10.36 was observed when AF488-labeled 3WJdB was used as imaging probe in place of 3WJdB–DFHBI-1T. **d**–**f** Cy3-labeled 3WJdB (green) was assembled with Waz aptamer, and colocalization with endocytic markers (red) was assessed after 1 h-incubation in NALM6 cells. **d** Representative confocal microscopy images of fixed and immunostained NALM6 cells show significant colocalization between Waz–3WJdB-Cy3 and Rab5 (early endosome marker). **e** A reduction of colocalization was found between Waz–3WJdB–Cy3 and Rab7 (late endosome marker). **f** NALM6 cells were co-incubated for 1 h with 0.5 μM AF488-labeled Tf and 0.5 μM Waz–3WJdB-Cy3 complex, then cells were fixed and imaged by confocal microscopy. A strong colocalization between Tf-AF488 and Waz–3WJdB-Cy3 was observed both in the cell periphery and perinuclear region of NALM6 cells. For all samples, Pearson's correlation coefficient was used to estimate the extent of colocalization between targeting and payload modules of the aptamer platform or between Waz–3WJdB–Cy3 and endocytic markers (see also Supplementary Fig. 14). Images are representative of two independent experiments. Scale bars: 5 μm

agreement with previous reports[52], potentially due to the interaction of this dye with intracellular components. Detection of green fluorescence from aptamer nanostructures in the same focal plane as the nuclear dye, rather than distributed around the edge of the cell, typically indicates that the payload has been internalized. To further test this, NALM6 cells were pre-incubated on ice to arrest endocytosis[53] prior to adding aptamer complexes. Waz- and C10.36-dependent cell staining was drastically reduced for both large and small payloads as measured by flow cytometry (Supplementary Fig. 13), indicating that an energy-driven process such as receptor-mediated endocytosis is responsible for cell uptake of the aptamer nanostructures. Overall, these results imply that the aptamer-dependent labeling of NALM6 cells observed by flow cytometry (Fig. 2) was due to internalized fluorescence. These findings are consistent with previous observations[44,54] that cell-internalizing aptamers typically possess a fast dissociation off-rate from their target receptors and do not remain associated with cells when endocytosis is arrested and cells are washed,

thereby leading to an overall reduction of aptamer-dependent cell staining.

**Aptamer nanostructure remains stably assembled in endosomes.** To assess whether RNA payloads remained annealed to cell-internalizing aptamers during internalization into NALM6 cells, dual-label colocalization studies were performed with Cy5 on the targeting module and AF488 or 3WJdB on the payload module. After incubating NALM6 cells with dual-labeled nanostructures for 1 h, fluorescent signal from the small RNA anti-tail strongly colocalized with either Waz or C10.36 (Pearson's correlation coefficient ~0.9) (Fig. 4a). The lower brightness and photostability of 3WJdB–DFHBI-1T[24,55] compared with AF488 and Cy5, and the intrinsic noise due to the unbound DFHBI-1T dye make 3WJdB fluorescence less than ideal for colocalization studies, which are most informative when fluorophores exhibit similar brightness and comparable background

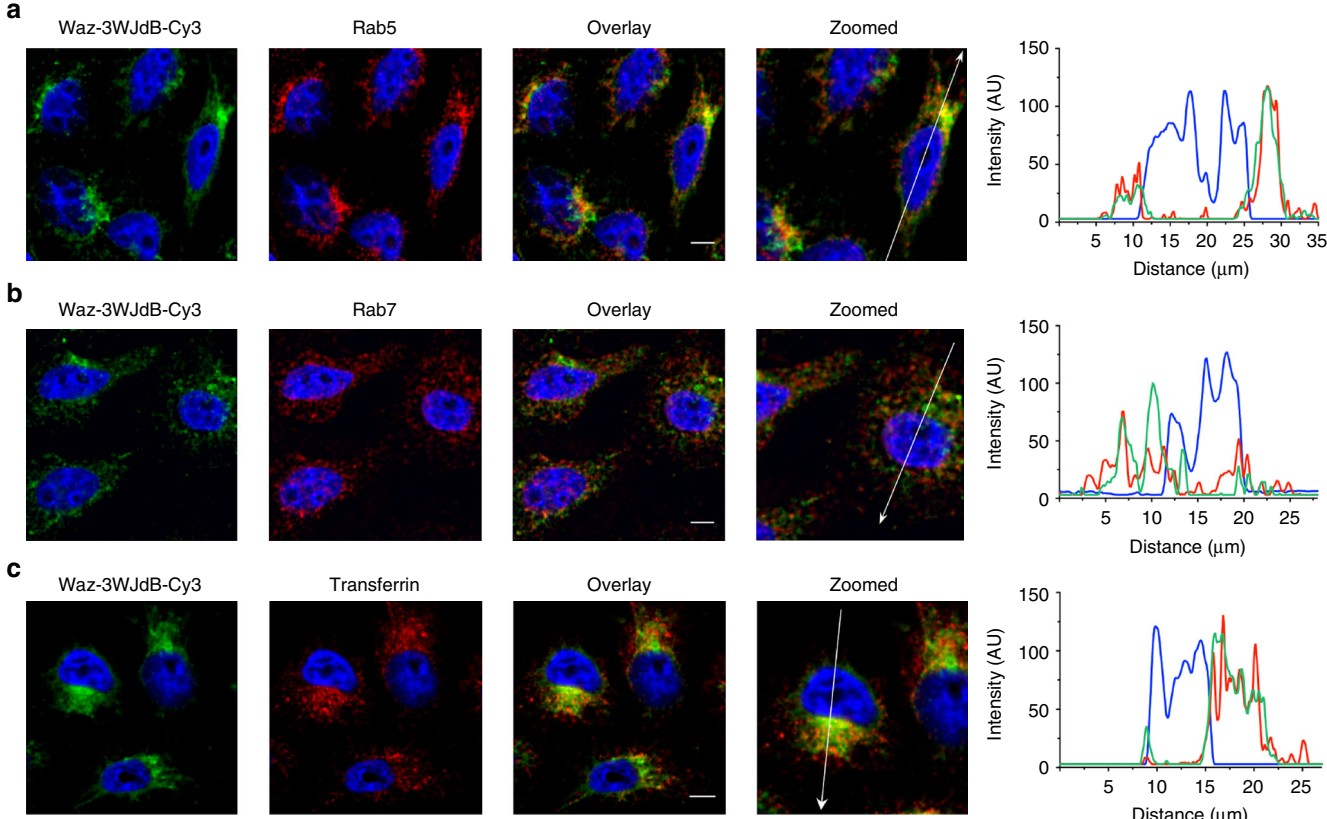

**Fig. 5** Aptamer nanostructures localize in endosomes of HeLa cells. Cy3-labeled 3WJdB (green) was assembled with 3-fold molar excess of Waz aptamer, and colocalization with endocytic markers (red) was assessed after 1 h-incubation in HeLa cells. **a** Representative confocal microscopy images of fixed and immunostained HeLa cells show significant colocalization between Waz–3WJdB-Cy3 and Rab5 (early endosome marker). **b** A reduction of colocalization was found between Waz–3WJdB-Cy3 and Rab7 (late endosome marker). **c** HeLa cells were co-incubated for 1 h with 0.5 μM AF488-labeled Tf and 0.5 μM Waz–3WJdB-Cy3 complex, then cells were fixed and finally imaged by confocal microscopy. A strong colocalization between Tf-AF488 and Waz–3WJdB-Cy3 was observed. A line-scan analysis (on the right of each set of images) was also performed to measure fluorescence intensity of Waz–3WJdB–Cy3 (green), Hoechst 33342 (blue), and endocytic marker (red) along a line drawn through the major axis of cells that intersects the nucleus. Fluorescence intensity was measured on this line for 10–15 μm from the nucleus on both sides. The zero of the distance scale refers to the beginning of the white arrow depicted in zoomed images. Line-scan analysis shows strong overlap of the Waz–3WJdB-Cy3 with both Rab5 and Tf intensity peaks, indicating significant colocalization. In contrast, line-scan analysis shows a small extent of overlap between Waz–3WJdB-Cy3 and Rab7 intensity peaks, suggesting only partial colocalization. For all samples, Pearson's correlation coefficient was also used to estimate the extent of colocalization between Waz–3WJdB-Cy3 and endocytic markers (see Supplementary Fig. 14). Images are representative of two independent experiments. Scale bars: 5 μm

fluorescence in the two channels[56]. Not surprisingly, when monitoring 3WJdB–DFHBI-1T fluorescence as a readout for this assay, colocalization of signals between targeting and payload aptamers was modest (Pearson's correlation coefficient ~0.4) in agreement with previous studies that reported how background noise makes the values of Pearson's correlation coefficient closer to 0 than they should be because the elevated background contributes to non-colocalized pixels[57] (Fig. 4b). However, replacing the small RNA payload with AF488-labeled 3WJdB with no further addition of DFHBI-1T, reduced the non-specific background and yielded strong and significant colocalization of the dye-labeled 3WJdB with both cell-targeting aptamers (Pearson's correlation coefficient ~0.8) (Fig. 4c). These results indicate that the delivery and payload modules are still assembled at early stages of endocytosis, even during delivery of large RNAs, highlighting the stability of the aptamer nanostructure in the endocytic vesicles.

To investigate sub-cellular localization of the large RNA payload upon aptamer-mediated delivery, we performed colocalization studies with endocytic markers Rab5, Rab7, and transferrin (Tf). 3WJdB was first labeled with Cy3 and then annealed to Waz. After 1 h incubation, confocal microscopy of

fixed and immunostained NALM6 cells showed that the majority of internalized Waz–3WJdB–Cy3 was within endosomes containing Rab5, a marker of early (sorting) endosomes[58] (Fig. 4d). The calculated Pearson's correlation coefficient was approximately 0.8 (Supplementary Fig. 14). Partial, albeit significant, colocalization (Pearson's correlation coefficient ~0.6, Supplementary Fig. 14) was also found between Waz–3WJdB–Cy3 and Rab7, a marker of maturing and late endosomes[58] (Fig. 4e). Waz binds a different epitope on hTfR than its natural ligand (Tf), thus enabling simultaneous binding by both ligands. NALM6 cells were co-incubated for 1 h with Tf-AF488 and Waz–3WJdB–Cy3. Strong colocalization between Tf and Waz–3WJdB (Pearson's correlation coefficient ~0.9, Supplementary Fig. 14) was detected both in the cell periphery and in perinuclear regions of NALM6 cells, indicating that Waz–3WJdB and Tf are transported toward the perinuclear region during the endosome maturation.

To further investigate sub-cellular localization of the aptamer nanostructures, we performed additional colocalization studies using HeLa cells, an hTfR-positive cell line that possess a more balanced ratio of nuclear-to-cytoplasmic area compared to the NALM6 B cell leukemia cell line. Figure 5 shows that Waz–3WJdB–Cy3 colocalized with Rab5 (Fig. 5a) and Tf-

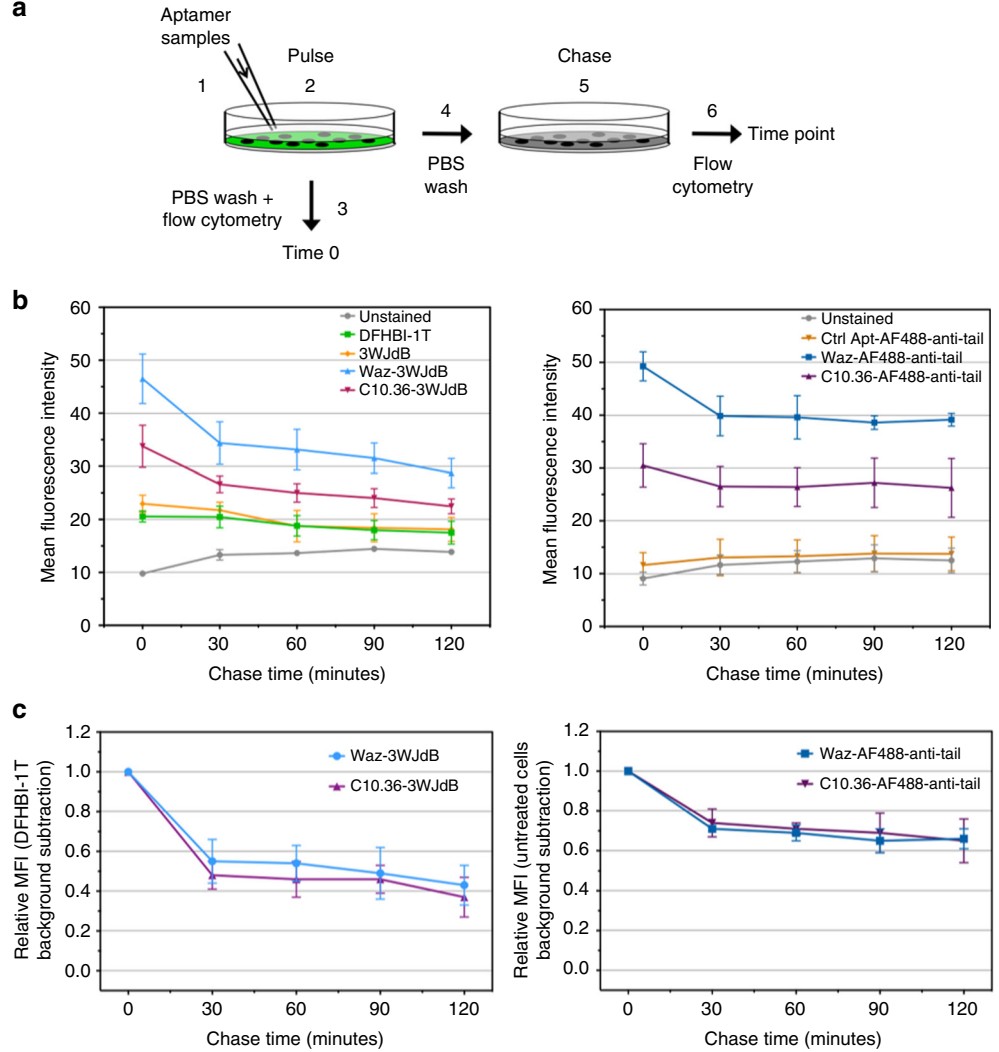

**Fig. 6** Persistence of 3WJdB fluorescence in leukemic B cells. **a** Schematic representation of the pulse-chase experiment performed on NALM6 cells. **b** Geometric mean fluorescence intensity of each time point is reported for both 3WJdB samples (on the left) and AF488-anti-tail samples (on the right). Values are the mean ± SD for three independent experiments. NALM6 cells incubated with 3WJdB samples were kept in medium supplemented with DFHBI-1T (20 μM) during the 1 h incubation (pulse phase), the entire chase phase (120 min), and the flow cytometry analysis. Representative flow cytometry curves for the pulse-chase experiment are shown in Supplementary Fig. 16. **c** Calculated relative mean fluorescence intensity (MFI) of Waz–3WJdB and C10.36–3WJdB (on the left) and Waz–AF488-anti-tail and C10.36–AF488-anti-tail (on the right). Relative MFI was calculated first by subtraction of background signal (DFHBI-1T only-treated cells or ctrl Apt-AF488-anti-tail) from all measurements. Then, for each sample, individual time points were normalized for the fluorescence at "time 0" of the same sample. Values are the mean ± SD for three independent experiments

AF488 (Fig. 5c) in the perinuclear region of HeLa cells (Pearson's correlation coefficients ≥ 0.7, Supplementary Fig. 14). Again, only a partial colocalization with Rab7 (Fig. 5b) was detected (Pearson's correlation coefficient ~0.5, Supplementary Fig. 14), suggesting that aptamer nanostructures are preferentially located in early/sorting endosomes and to a lesser extent in maturing/late endosomes. Line-scan analysis[59] of fluorescence intensity along the major axis of the cells (see intensity plots in Fig. 5) further confirmed the extent of colocalization of Waz–3WJdB–Cy3 (green) with endocytic markers (red). Strong overlap of Waz–3WJdB with both Rab5 and Tf intensity peaks was observed, indicating high levels of colocalization. In contrast, line-scan analysis shows only a small extent of overlap between Waz–3WJdB and Rab7 intensity peaks, suggesting only partial colocalization.

Nearly identical sub-cellular localization of Waz–3WJdB–Cy3 was detected in a second hTfR-positive cancer cell line (MDA-MB-231) with a balanced ratio of nuclear-to-cytoplasmic area.

Confocal microscopy analysis of MDA-MB-231 cells (Supplementary Fig. 15) again displayed a punctate pattern of Waz–3WJdB that colocalized with Rab5- and Tf-positive endosomes, and to a lesser extent with Rab7-containing vesicles (see also calculated Pearson's correlation coefficients in Supplementary Fig. 14). Overall, these results strongly indicate that aptamer nanostructures primarily localize to early/sorting endosomes upon receptor-mediated endocytosis.

**Endocytosed 3WJdB persists upon aptamer-mediated delivery.** Pulse-chase analysis was performed to monitor persistence and intracellular processing of endocytosed large RNA payload upon aptamer-mediated delivery. NALM6 cells were first loaded with aptamer nanostructures or their respective controls (step 1 in Fig. 6a). After 1 h incubation with nanostructures and DFHBI-1T ("pulse", step 2), cells were either washed and analyzed by flow cytometry ("time 0", steps 3 and 4) or kept in aptamer-free medium for various times ("chase", step 5) prior to measuring cell

staining by flow cytometry (step 6). Interestingly, the precipitous drop of 3WJdB fluorescence in the first 30 min was followed by a much more gradual decline over time (Fig. 6b, c and Supplementary Fig. 16). Even after a 120-min chase, ~40% of the initial 3WJdB fluorescence was still present, indicating that a significant fraction of the 3WJdB that had been taken up during the pulse still retained productive folding (Fig. 6c). Notably, reduction of 3WJdB fluorescence during the chase time followed a similar trend irrespective of whether delivery was accomplished with Waz or C10.36. In contrast, the weak staining of NALM6 cells that resulted from non-specific uptake of free 3WJdB alone—lacking the targeting modules—fully disappeared after 30-min chase (Fig. 6b). When fluorogenic RNA was replaced with AF488-anti-tail as payload, fluorescence again decreased in the first 30 min and then declined much more slowly (Fig. 6b, c and Supplementary Fig. 16). This result was somewhat expected, given that dye fluorescence is not affected by folding alterations or enzymatic degradation. This pulse-chase analysis, which monitors fluorescence from functional 3WJdB as a function of time, demonstrates that this large RNA payload persists in a full-folded and functional form within the intracellular compartment, potentially increasing opportunities for other RNA payloads to engage endosomal targets or to escape from the endosomes into the cytosol.

## Discussion

We developed and characterized a modular nanostructure to perform aptamer-mediated targeted delivery of large functional RNAs. This platform exploits cell-internalizing aptamers to accomplish cell-specific targeting and delivery, along with the fluorescence properties of RNA mimics of GFP as surrogates for other large RNA-based payloads. The fluorogenic properties of the enhanced Broccoli aptamer variants were used with a dual function: (i) to assess retention and persistence of aptamer payload folding when assembled into a more complex RNA structure both in vitro and in live cells, and (ii) to demonstrate aptamer-mediated targeted delivery of large functional RNAs (at least 244 nt; ~80 kDa) in cancer cells.

Self-assembly of aptamer nanostructures was performed using excess targeting aptamer over the payload to minimize the fraction of free fluorogenic RNA aptamer. This strategy has two benefits. First, since annealing of targeting and payload modules occurs with high efficiency (Fig. 1 and Supplementary Fig. 1), fluorescence of the RNA payload emerges directly from the assembled delivery platform. Second, only short incubation times and low concentrations of payload aptamer were used during the incubation with the cells. This latter is a critical condition to minimize non-specific cell uptake of non-internalizing oligonucleotides, which occurs via non-specific mechanisms of endocytosis (such as macropinocytosis)[48] at relatively high concentration (≥1.0 μM) of oligos. Minimization of non-specific binding and uptake is essential, especially when evaluating the targeting and internalization properties of candidate targeting ligands.

Flow cytometry analysis demonstrated that both cell-internalizing aptamers (Waz and C10.36) were able to deliver two different large, functional RNA (3WJdB, 176 nt; and 3WJtriB, 244 nt) to target cancer cell lines, such as NALM6 (Fig. 2 and Supplementary Fig. 5), but not to control cells (Fig. 2 and Supplementary Figs. 6 and 7). These results demonstrate modularity of the design of our cell-internalizing aptamer nanostructure, which enables plug-and-play engineering by swapping cell-targeting aptamer or RNA payload without losing activity. This offers the potential to target virtually any cell line of interest by simply incorporating the proper cell-targeting aptamer in our delivery platform.

Confocal microscopy imaging (Fig. 3 and Supplementary Fig. 12) confirmed that internalized RNA payload is the major contribution to total cell labeling that is measured by flow cytometry. Further experiments performed on cells kept on ice (Supplementary Fig. 13) suggested that an energy-dependent mechanism of endocytosis is responsible for uptake of aptamer nanostructures in NALM6 cells. Upon internalization, both small (AF488-anti-tail) and large (3WJdB) RNA displayed a punctate, perinuclear pattern of NALM6 cells, which is consistent with endosomal trafficking vesicles (Figs. 3 and 4). Colocalization studies were performed using three different cancer cell lines (NALM6, HeLa, and MDA-MB-231) to investigate sub-cellular localizations of aptamer nanostructures. 3WJdB strongly colocalized with Rab5 and Tf containing vesicles, indicating that the primary localization of this large RNA payload after 1 h incubation is early/sorting endosomes (Figs. 4, 5, and Supplementary Figs. 14 and 15). These colocalization results, together with the reduced cellular uptake when cells were incubated on ice, indicate that the aptamer nanostructures are internalized via receptor-mediated endocytosis and are transported from the cell periphery toward the perinuclear region following endosome maturation. The moderate colocalization between 3WJdB and Rab7 suggests that a fraction of this RNA payload aptamer is likely located in maturing endosomes (both Rab5- and Rab7-positive vesicles)[58] or late endosome (Rab7-containing endocytic vesicles). Furthermore, strong colocalization observed between RNA payloads and cell-internalizing aptamers in NALM6 cells (Fig. 4a–c) indicate that the aptamer nanostructure remains stably assembled after 1 h incubation in the endosomes of target cells.

The extent of 3WJdB and AF488-anti-tail delivered by Waz positively correlated with the levels of hTfR on target cancer cells measured with an anti-hTfR antibody (Supplementary Figs. 9 and 10), further confirming that the binding of a specific target receptor is responsible for the aptamer-mediated delivery of RNA payloads into target cells. Such confirmation is critical for studies that investigate targeting properties of aptamers to avoid false positives and erroneous interpretation.

The persistent retention of 3WJdB signal in live cells has intriguing implications for the potential existence of long-lived reservoirs for delivered nucleic acids. Specifically, using a pulse-chase study, a significant fraction of functional 3WJdB (~40%) was still detected after 120-min chase (Fig. 6c). Because fluorescence signal requires folded, fully-functional Broccoli aptamer modules, this assay reports on the fraction of fully-intact payload RNA. The longevity of RNAs inside the cells can vary depending on their structures or association with proteins to form RNP complexes[60]. Several reports have described that siRNAs delivered via endocytosis possess surprisingly long intracellular persistence. Intact 2′OH-siRNAs were found inside cells even after 3 days of incubation; they appeared to be released in a biphasic manner, with an initial burst release from endosomes to cytosol, followed by a slower, albeit continuous, release from endosomal reservoirs[61]. The long persistence of siRNAs contribute to the efficiency and duration of RNAi in cell culture and in vivo, leading to sustained gene silencing[61,62]. Analogously, our findings show that a significant fraction of 3WJdB persists inside the cells in a functional status for the entire duration of our chase step (2 h). The long intracellular persistence of 3WJdB could be associated with the complex vesicle trafficking that occurs during endosome maturation. Our colocalization studies indicate that, after 1 h incubation, 3WJdB is sorted in different endosomes (Rab5- and to a lesser extent Rab7-positive vesicles). Previous reports described that early endosomes possess different trafficking and kinetics of maturation[58]. This could affect both intracellular localization and integrity of RNAs located in endocytic vesicles. The population of early endosomes that matures

quickly toward late endosomes and ultimately to lysosomes could lead to a fast degradation of RNA payloads, unless endosomal escape events occur. In contrast, the relatively static population of early endosomes with a slow kinetic of maturation could traffic RNAs in non-degradative vesicles for a long time, thus acting as endosomal reservoirs[61]. These slowly maturing early endosomes and their trafficking could be responsible for the long intracellular persistence of 3WJdB.

Overall, these results provide a basis for further investigations related to the ability of this or other large RNAs to escape from endosomes and access the cytosol. Recent findings suggested that enhancing the residence time in the endosomal compartment can extend the window of time for a more efficient escape of payloads into the cytosol (also known as "window of opportunity") before their ultimate degradation in the lysosomes[8,63]. Therefore, RNA payloads that still remain in their proper folding and are not immediately digested in endocytic vesicles could have higher chances to escape by exploiting this extended "window of opportunity."

This work shows that we can deliver functional RNA into cells and that it retains its folding and function once it gets there. Although endosomal escape is an essential step in many therapeutically relevant applications, such as RNAi-mediated gene silencing, other strategies aim at targeting endosomes directly[64,65]. The endosomal compartment is not a simple transient environment for the degradation or recycling of cell-surface receptors and uptake of nutrients. Rather, this complex organelle system also represents an essential site of signal transduction mediated by different classes of receptors, such as receptor tyrosine kinases (RTKs), G protein-coupled receptors (GPCRs), and Toll-like receptors (TLRs)[64]. Particularly, mutations affecting some of these receptors or their corresponding adaptor proteins are involved in the signal transduction that leads to tumorigenesis in several forms of cancers[64,66]. Given that aptamers are very good at binding membrane proteins, the door is now open for targeting proteins that reside within intracellular vesicles so as to engage endosomal biology and modulate cellular outcomes at the endosomal level.

## Methods

**Materials**. All chemicals were purchased from Sigma-Aldrich (St Louis, MO, USA) unless otherwise noted. Cell culture products were purchased from Gibco BRL/Life Technologies (Gaithersburg, MD, USA). APC-labeled-anti-CD71 monoclonal antibody (clone OKT9) was purchased from eBioscience (San Diego, CA, USA).

**Cell culture**. NALM6 human leukemia cells and MDA-MB-231 breast cancer cells were purchased from the American Type Culture Collection (ATCC, Manassas, VA). MEC-1 and MEC-2 human leukemia cells were purchased from Leibniz Institute DSMZ-German Collection of Microorganisms and Cell Cultures (DSMZ, Germany). HeLa cells were gifted by Dr. Christian Lorson, University of Missouri-Columbia. SP1 cells[67] were gifted by Dr. Michael Farrar, University of Minnesota. Canine lymphoma B cell line, CLL17, and canine leukemia T cell line, CLGL-90, were gifted by Dr. Sandra M. Bechtel, University of Missouri-Columbia. HeLa cells were maintained in DMEM containing 1 mM sodium pyruvate, 2 mM L-glutamine, and 10% FBS; SP1 cells were maintained in Opti-MEM-reduced serum media supplemented with 1 ng/ml of Interleukin-7 (IL-7), 2 mM L-glutamine, and 4% FBS; and all other cell lines were maintained in RPMI 1640 medium containing 1 mM sodium pyruvate, 2 mM L-glutamine, supplemented with 10% FBS. Cells were maintained at 37 °C in a humidified 5% CO$_2$ atmosphere.

**Design of the aptamer nanostructure and NUPACK analysis**. All DNA and RNA sequences used in this study are reported in Supplementary Table 1. Secondary structure predictions of aptamers were calculated using the NUPACK web application[46] (http://www.nupack.org/) at default settings at 37 °C. The in-silico design of the aptamer nanostructure was assessed by NUPACK using the hybridization tool. During the calculation of annealing, the two input strands (targeting and payload) were constrained to a maximum complex size of 2 and at a concentration of 0.5 μM each. A depiction of 3D aptamer structures was obtained by exploiting the utilities of the NUPACK software.

The secondary structure of the 3WJ architecture is correctly predicted pointing to a reliable conformation for the annealing region (also called connector

domain)[29]. Although the quadruplex motifs and several non-canonical base pairings in the dB and C10.36 aptamers are not accurately reproduced in the NUPACK predictions (Fig. 1b), this approach can identify designs that preferentially form stable alternative (inactive) folds for certain annealed complexes so that those candidate sequences can be eliminated. NUPACK-predicted secondary structure of 3WJdB and 3WJtriB, shown in Supplementary Fig. 4, were depicted using VARNA software[68].

**DNA templates, list of primers and RNA transcription**. All DNA oligos were purchased from Integrated DNA Technologies (IDT, Coralville, IA, USA) and resuspended in the appropriate volume of TE buffer (10 mM Tris–HCl, pH 8.0, 1 mM EDTA) to reach a stock concentration of 100 μM. RNA anti-tail and DNA aptamer C10.36 with 3′ extension were purchased with a 5′-amino group attached by a C-6 alkyl chain (5′ Amino Modifier C6), which was exploited for dye-labeling reactions.

List of primers used in this study:
Waz forward primer: 5′-TAATACGACTCACTATAGGGGTTCTACGAT-3′
Waz reverse primer: 5′-TCGTCGTCGTCGTCGTCGTCGGGGAACTGCCAG-3′
3WJdB forward primer: 5′-GCCTAATACGACTCACTATAGGA-3′
3WJdB reverse primer: 5′-CGACGACGACGACGACGACGACCCACATACACATGGCAAGA-3′
3WJtriB forward primer: 5′-GCCTAATACGACTCACTATAGGA-3′
3WJtriB reverse primer: 5′-CGACGACGACGACGACGACGAATCCGCATCATCTATCT-3′

All RNA aptamers were generated via in vitro run-off transcription. The corresponding DNA templates were first ligated and PCR-amplified with primers that appended a T7 promoter. 3WJdB transcription was performed as previously described[29]. Waz aptamer was transcribed by run-off transcription reaction (overnight at 37 °C) using a recombinant mutant form of T7 RNA polymerase (Y639F mutant), in vitro transcription buffer (50 mM Tris–HCl, pH 7.5, 15 mM MgCl$_2$, 5 mM DTT, 4% w/v PEG4000, and 2 mM spermidine), and 2 mM of each ATP, GTP, 2′-fluoro-modified CTP, and 2′-fluoro-modified UTP (TriLink Biotechnologies, San Diego, CA, USA). All RNAs were purified through denaturing polyacrylamide gel electrophoresis (0.75 mm 6 or 8% TBE-PAGE, 8 M urea) and bands corresponding to the expected product size were gel extracted and eluted while tumbling overnight in 300 mM sodium acetate, pH 5.4. Eluates were ethanol precipitated, resuspended in TE buffer, and stored at −20 °C until further use.

**Aptamer annealing and fluorescence measurements**. In vitro assembly reactions were prepared on ice in "cell-binding buffer" (Dulbecco's phosphate buffered saline, DPBS, containing 40 mM HEPES, pH 7.5, 100 mM KCl, 5 mM MgCl$_2$, and 20 μM DFHBI-1T) using an excess of targeting aptamer over the payload (3:1 molar ratio of targeting:payload, 1.5 μM targeting:0.5 μM payload aptamer). DFHBI-1T (Lucerna Technologies, Brooklyn, NY, USA) was added to the cell-binding buffer only for samples containing fluorogenic RNA aptamers (3WJdB or 3WJtriB). For thermal renaturation, samples were transferred into a preheated aluminum insert within a dry heat block set to 90 °C. Samples were kept at 90 °C for 1 min, and then the aluminum insert was removed from the block heater and placed on the workbench to cool down to 37 °C before assessing annealing and fluorescence. An aliquot of each sample (50 μl) was then transferred into a clear, flat-bottom 96-well plate and measured for fluorescence using an EnSpire Multimode plate reader (PerkinElmer, Waltham, MA, USA) at room temperature. Fluorescence measurements were performed under two different in vitro setting conditions: (i) preferential Broccoli aptamer settings (λex: 472 nm; λem = 492–600 nm) and (ii) standard flow cytometry settings (λex: 488 nm; λem = 508–600 nm). Fluorescence of each sample was normalized for to 3WJdB fluorescence at each setting, and normalized values were used to calculate means and standard deviations.

**Dual staining native gel shift assay**. Native 6% polyacrylamide gels were prepared to check the electrophoretic mobility shift relative to the formation of aptamer–aptamer complexes. Aptamer samples were annealed in the same cell-binding buffer described above, but without the addition of DFHBI-1T. Upon annealing, an aliquot of each sample containing 10 pmol of fluorogenic RNA aptamer (3WJdB or 3WJtriB) was loaded onto a 0.75 mm 6% native TBE polyacrylamide gel in a final volume of 50 μl with 20% glycerol. Approximately 1 h 30 min after electrophoresis at 10 W at 4 °C, the gel was stained according to an in-gel imaging protocol[45]. In brief, the gel was stained with 5 μM of DFHBI-1T at room temperature for 15 min and then imaged using a Typhoon FLA 9000 (GE Healthcare) with Alexa Fluor 488 settings (473 nm laser excitation, Y520 emission filter). A de-staining step was then performed on the gel with two washes in ultrapure Milli-Q water (EMD Millipore) for 5 min each, followed by a 5-min incubation in ethidium bromide at 0.5 μg/ml. The gel was then reimaged on a Typhoon FLA 9000 (GE Healthcare) using ethidium bromide settings (532 nm laser excitation, O580 emission filter). Densitometry analysis of gel bands was performed using Fiji[69]. After linear contrast adjustment, the intensity value of each band was estimated. The hybridization yield for the annealed complex was calculated according to the following formula: [(annealed complex)/(free 3WJdB + annealed complex)].

**5′-modification of RNA aptamers and dye-labeling reactions**. RNA aptamers (3WJdB and Waz) were generated via in vitro run-off transcription in the presence of an excess of guanosine 5′-monophosphate (GMP) over guanosine-5′-triphosphate (GTP) (molar ratio 4:1 GMP:GTP). All RNAs were purified through denaturing polyacrylamide gel electrophoresis as described above. Eluates were ethanol precipitated and resuspended in "reaction buffer" (DPBS containing 10 mM sodium phosphate, 0.15 M NaCl, and 10 mM EDTA, pH 7.2). To generate 5′-NH$_2$-labeled RNA, 1 nmol of RNA dissolved in reaction buffer was mixed with 1.25 mg (6.52 μmol) of 1-ethyl-3-(3-dimethylaminopropyl) carbodiimide hydrochloride (EDC), 20 μl of ethylenediamine (0.25 M in 0.1 M imidazole, pH 6), and 10 μl of imidazole (0.45 M, pH 6). The mixture was incubated at room temperature for 6 h. The non-reacted EDC, ethylenediamine, and imidazole were removed using a spin desalting column (Amicon Ultra-0.5mL, ultracel-3membrane, 3K MWCO, Millipore, Burlington, MA, USA) and reaction buffer for the washing steps. The amino residues at 5′-end of RNA (3WJdB, Waz) and DNA (C10.36) sequences were conjugated to N-hydroxysuccinimidyl ester (NHS ester) fluorophore derivatives according to a labeling protocol previously described[53]. 3WJdB was labeled with AF488-NHS Ester (Thermo Fisher Scientific, Waltham, MA, USA) whereas Waz and C10.36 were labeled with Cy5-NHS ester (Lumiprobe, Hunt Valley, MD, USA). Analytical evaluation of labeling reactions and purification of dye-labeled aptamers were performed using reverse-phase HPLC (RP-HPLC). An Agilent 1100 series instrument with an Agilent Zorbax Eclipse XDB-C18(4.6 × 150 mm$^2$) were employed for RP-HPLC analysis. We used a flow rate of 1 ml/min with triethylammonium acetate (TEAA)/acetonitrile (ACN) buffer system (solvent A: 100 mM TEAA, pH 7; solvent B: 100% ACN). Labeled aptamers were separated from the unlabeled fraction and purified using a linear gradient from 5 to 30% ACN over 20 min. A further linear gradient was applied from 30 to 60% ACN over 10 min to purge the column of unreacted fluorophore. Purified dye-labeled aptamers were concentrated and purified by the residual fraction of ACN using spin desalting column (Amicon Ultra-0.5mL, 3K MWCO). For each column wash, 300 μl of TE buffer was added to the spin column followed by centrifugation at 8000×g for 20 min at 4 °C. Each sample was concentrated to ~50 μl. Dye-to-aptamer ratio was evaluated from absorbance measured at 260 nm (for DNAs and RNAs) and 494 nm (for AF488) or 646 nm (for Cy5) on a NanoDrop 1000 spectrophotometer (Thermo Scientific). All dye-labeled aptamers exhibited a dye-to-aptamer ratio of 1 within experimental error. To label transferrin protein, 1 nmol of Tf (apo-transferrin, Sigma-Aldrich) dissolved in reaction buffer was incubated with 10-fold molar excess of AF488-NHS ester in DPBS at room temperature for 2 h. AF488-labeled Tf (Tf-AF488) was separated and purified by the fraction of free fluorophore using spin desalting column (Amicon Ultra-0.5mL, 30K MWCO). Approximately 8–10 column washes were performed with 300 μl of DPBS followed by centrifugations at 8000×g for 20 min at 4 °C. A dye-to-protein ratio of ~2.5 was measured from absorbance at 280 nm (for Tf) and 494 nm (for AF488) on a NanoDrop 1000 spectrophotometer.

**3′-labeling of 3WJdB with Cy3-hydrazide**. For colocalization studies between 3WJdB and endocytic markers (Rab5, Rab7, and Tf) in cancer cell lines, 3WJdB was labeled at its 3′-end with Cy3-hydrazide (Lumiprobe, MD, USA) exploiting a more rapid and efficient strategy of RNA labeling than the aforementioned 5′-modification. The labeling protocol was modified from a previous report[70]. First, 3WJdB was generated via in vitro run-off transcription, purified by PAGE, ethanol precipitated, and resuspended in DPBS. To oxidize the 2′,3′ diols of the RNA to aldehydes, 1 nmol of 3WJdB was mixed with NaIO$_4$ (10 mM final concentration) in 300 mM sodium acetate pH 5.4 for 2 h at 4 °C in the dark. The excess of NaIO$_4$ was then removed using spin desalting column (Amicon Ultra-0.5mL, 3K MWCO) and ethanol precipitation of the RNA. 3WJdB was then resuspended in DPBS and mixed with 20-fold molar excess of Cy3-hydrazide in 300 mM sodium acetate, pH 5.4, overnight at 4 °C in the dark (or alternatively 2 h at room temperature). Analytical evaluation of labeling reactions and purification of dye-labeled aptamers were performed using RP-HPLC as described above. Coupling efficiency was approximately 60%. Dye-to-aptamer ratio was evaluated from absorbance measured at 260 nm (RNA) and 540 nm (for Cy3) using a NanoDrop as described above.

**Flow cytometry assay**. Assembly of aptamer nanostructures in cell-binding buffer was performed as described above (using 1.5 μM C10.36 or Waz and 0.5 μM 3WJdB or AF488-anti-tail). DFHBI-1T (20 μM) was included in cell-binding buffer when 3WJdB was used as payload. Upon assembly, competitors for non-specific binding sites, such as yeast tRNA and salmon sperm DNA (ssDNA), were added to each aptamer sample at a final concentration of 0.5 mg/ml each. To minimize non-specific binding and uptake of the aptamer nanostructures, stringently controlled conditions were used: (i) short incubation time (1 h), (ii) relatively low concentration of aptamer–aptamer complexes (~0.5 μM), and (iii) co-incubation with 1 mg/ml of tRNA + ssDNA (0.5 mg/ml each). Cancer cells (5 × 10$^5$ cells/sample) were incubated at 37 °C in 100 μl of aptamer samples, in cell-binding buffer containing DFHBI-1T 20 μM without aptamer complexes, or in cell-binding buffer only without DFHBI-1T 20 μM. Upon 1 h incubation, cells were washed twice with 200 μl DPBS, followed by final resuspension in cell-binding buffer (±DFHBI-1T 20 μM depending on the sample used). Flow cytometry was performed on a BD FACSCalibur (BD Biosciences, San Jose, CA, USA) and 5 × 10$^4$ events were

analyzed and processed using FlowJo Software (Treestar, Ashland, OR, USA). Viable cells were gated and analyzed, while dead cells were identified by the forward scatter plot and excluded from the analysis.

To assess whether the internalization of the aptamer nanostructures was due to an energy-dependent process, NALM6 cells were pre-incubated in serum-free medium (RPMI buffer) on ice for 10 min to arrest endocytosis, and then incubated with the aptamer samples, still on ice for 1 h in cell-binding buffer. The same protocol described above was used to assess cell staining by flow cytometry.

For the pulse/chase experiment, upon 1 h-incubation with aptamer samples, NALM6 cells were washed twice with DPBS to remove unbound aptamers and incubated in aptamer-free medium (either cell-binding buffer or cell-binding buffer containing 10% FBS) for additional 120 min (chase period). Flow cytometry analysis was performed at each time point as described above. Fluorescence changes over the time were not significantly affected by the presence of 10% FBS in the cell culture medium. DFHBI-1T (20 μM) was included in cell-binding buffer only when 3WJdB was used as payload. To calculate the relative fluorescence shown in Fig. 5, background signal (DFHBI-1T only-treated cells for 3WJdB samples, and ctrl Apt-AF488-anti-tail for AF488 samples) was subtracted from all measurements, and then, for each sample, individual time points were normalized for time 0 of the same sample. Normalized values were used to calculate means and standard deviations.

Cell staining with the anti-CD71 antibody was used to assess the level of Tf receptor surface expression. Cells (5 × 10$^5$ cells/sample) were incubated with 0.06 μg of clone OKT9 anti-CD71 antibody for 30 min. Cells were washed with 200 μl of DPBS and flow cytometry was performed as described above.

**Confocal laser microscopy imaging and image analysis**. Aptamer nanostructures for fluorescence microscopy were prepared as described above. NALM6 cells (1 × 10$^6$) were incubated at 37 °C for 1 h with the aptamer samples. After incubation, NALM6 cells were washed with DPBS, transferred to coverslips, and fixed with 1% paraformaldehyde in DPBS (45 min at room temperature). After fixation, cells were washed with DPBS and the nucleus was stained for 5 min at room temperature using 10 μg/ml of Hoechst 33342 (Thermo Fisher Scientific). Coverslips were mounted overnight on glass microscope slides using FluorSave$^{TM}$ reagent (Calbiochem, San Diego, CA, USA). Cells were then imaged with a confocal laser microscope (Leica TCS SP8) equipped with a hybrid detector (commercially, HyD$^{TM}$) and a highly tunable pulsed white-light laser (commercially, WLL) (Leica Microsystemes, Wetzlar, Germany). The Leica Application Suite X (LAS X) was used as a software platform. Images were acquired using the HC PL APO 63 × 1.4 Oil CS2 as objective lens, under sequential mode (scan mode: xyz; scan speed: 600 Hz) to avoid crosstalk (or bleed-through) between detectors, with an image size of either 512 × 512 or 1024 × 1024 pixels. The pinhole aperture was set to 1.0 Airy. Tailored conditions relative to excitation and emission wavelength were set up for each fluorescent probe. 3WJdB fluorescence was measured using 482-nm excitation laser from the white-light laser and the hybrid detector at 495–550 nm wavelength region. AF488 fluorescence was measured using 495-nm excitation laser from the white-light laser and the hybrid detector at 505–550 nm wavelength region. Cy5 fluorescence was measured using 647-nm excitation laser from the white-light laser and the hybrid detector at 660–690 nm wavelength region. Hoechst 33342 fluorescence was measured using 405-nm excitation laser from a diode laser (PicoQuant) and the hybrid detector at 415–440 nm wavelength region. Bright field images were captured using a photomultiplier tube (PMT) for transmittance (Scan-BF mode of PMT trans in LAS X software). All data were exported and processed into Fiji for linear adjustment of the brightness and contrast. The extent of colocalization was identified by Coloc 2 plugin of Fiji (ImageJ). Pearson's correlation coefficient values range from −1 to +1, where −1 is total negative correlation, 0 is no correlation, and +1 is total positive correlation. We averaged over five fields of view for each of the biological replicates. Line-scan analysis was performed on representative confocal microscopy images of HeLa cells using Fiji to qualitatively visualize fluorescence overlap. During this analysis, a line was drawn manually through the longest part of the cell that intersects the nucleus. For each image, fluorescence intensities of 3WJdB–Cy3, Hoechst 33342, and the corresponding endocytic marker (Rab5, Rab7, or Tf) were measured on this line for 10–15 μm from the nucleus on both sides. Plot profiles were acquired in Fiji using the same lines.

**Immunofluorescence microscopy of Rab5 and Rab7**. Waz–3WJdB–Cy3 (1.5 μM Waz and 0.5 μM 3WJdB–Cy3) was prepared as described above. 1 × 10$^6$ suspension B cell leukemia cells (NALM6) were incubated at 37 °C for 1 h with the aptamer samples (following the same protocol described in "Flow Cytometry Assay"). After incubation, NALM6 cells were washed with DPBS, transferred to coverslips, fixed with BD Cytofix$^{TM}$ (30 min at 4 °C) and permeabilized with BDCytoperm$^{TM}$ (10 min at room temperature) via the BD Cytofix/Cytoperm™ Fixation/Permeabilization Solution Kit, BD Biosciences. In the case of adherent cancer cell lines (HeLa and MDA-MB-231), cells were seeded at 5 × 10$^5$ cells per well in an 8-well Lab-Tek chambered coverglass (Nunc$^{TM}$ Lab-Tek$^{TM}$ chamber slide, Thermo Scientific). The following day, 180 μl of fresh serum-free medium containing Waz–3WJdB–Cy3 (1.5 μM Waz and 0.5 μM 3WJdB–Cy3), and a mix of tRNA + ssDNA (0.5 mg/ml each) in cell-binding buffer was added to cells. After 1 h incubation, cells were washed with DPBS, fixed and permeabilized directly on Lab-

Tek chambered cover glass as described above. After fixation and permeabilization, all cells (NALM6, HeLa, and MDA-MB-231) were incubated with a DPBS solution containing 3% BSA as blocking agent (1 h at room temperature). Cells were then washed with BDCytoperm™ and incubated with the primary antibody for 1 h at room temperature in BDCytoperm™ using the following antibodies: Rab5 (rabbit, 1:500, C8B1 mAb 3547, Cell Signaling Technology, MA, USA), Rab7 (rabbit, 1:500, D95F2 XP mAb 9367, Cell Signaling Technology, MA, USA). Cells were washed with BDCytoperm™ and incubated with Alexa Fluor 647-labeled secondary antibody (1:200, 1 h, dark, room temperature). Next, cells were washed with BDCytoperm™ and nuclei were stained for 10 min at room temperature using 10 µg/ml of Hoechst 33342. Cells were mounted overnight using FluorSave™ reagent, and finally imaged to assess sub-cellular localization of either Rab5 or Rab7 using the same confocal microscopy settings described above. Cy3 fluorescence was measured using 550-nm excitation laser from the white-light laser and the hybrid detector at 560–600 nm wavelength region. AF647 fluorescence was measured using the same settings used for Cy5 and described above. In colocalization studies between Waz–3WJdB and Tf, cancer cell lines were co-incubated for 1 h with serum-free medium containing Waz–3WJdB–Cy3 complex (1.5 µM Waz and 0.5 µM 3WJdB–Cy3), 0.5 µM AF488-labeled Tf, and a mix of tRNA + ssDNA (0.5 mg/ml each) in cell-binding buffer. Then, cells were fixed and imaged by confocal microscopy following the protocol described above.

**Statistical analysis**. Experimental $n$ values can be found in the figure legends. All data are expressed as mean ± SD. Statistical analysis for comparing multiple groups was performed using one-way analysis of variance (ANOVA) with a Tukey's multiple-comparisons test. Student's $t$-test (two-tailed) was applied to determine $p$ values relative to the correlation between Waz-dependent cell staining and TfR expression as measured by anti-CD71 Ab. The specific statistical methods applied and descriptions of replicates can be found in the figure legends. A value of $p < 0.05$ was considered statistically significant. Analyses were performed with Prism 6 (Graph Pad Software, San Diego, CA). Differences are labeled as follows: ns for not significant, * for $p < 0.05$; ** for $p < 0.01$; *** for $p < 0.001$; and **** for $p < 0.0001$.

**Data availability**. The data sets within the article and supplementary files generated during the current study are available from the authors upon request.

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

## Acknowledgements

This work was supported by National Institutes of Health grants (R01AI074389 and R21AI121938) to D.H.B., and by the University of Missouri. The authors would like to thank Dr. Alexander Jurkevich (MU cytology core) for assistance with confocal microscopy analysis.

## Author contributions

D.P., K.K.A., and D.H.B. conceived and designed this work. D.P., L.N.C., K.D.T., K.K.A., M.J.L., M.A.D., and D.H.B. designed the experiments. D.P. and K.D.T. performed all enzymatic oligo synthesis, electrophoretic gel analyses, and in vitro fluorescence measurements. D.P. and L.N.C. performed confocal imaging and flow cytometry studies. L.N.C. performed cell-based experiments. D.P. performed dye-labeling of oligos and HPLC purification. All authors discussed the results and commented on the manuscripts. D.P. and D.H.B. wrote the manuscript.

## Additional information

**Competing interests:** The authors declare no competing interests.

