## [Peer Review File · Nature Communications]

Reviewers' comments:

Reviewer #1 (Remarks to the Author):

The manuscript by Porciani et al. describes the application of aptamer nucleic acids that target specific cell surface receptors to deliver large non-coding RNAs to mammalian tissue culture cells. Previously, other groups have employed similar approaches to deliver small oligonucleotides, but this is, to my knowledge, the first application of this technology for delivering larger (> 100 nt) RNAs. The authors use dimeric and trimeric forms of the fluorogenic RNA broccoli as a cargo, and take advantage of the fluorescent properties of broccoli bound to its chromophore to report on cell internalization. The work appears technically sound in general, and I think it will be of interest to those in the RNA delivery and RNA therapeutics fields. I think the manuscript is topical and generally appropriate for this journal, but I have a couple of concerns that the authors may wish to address.

1. As the authors point out, Broccoli and its relatives are structured around a G-quadruplex motif. These are known to be generally very stable secondary structure motifs, and to my knowledge all characterized cellular helicases that destabilize G-quadruplexes require a 3' single-stranded extension immediately adjacent to the G-quadruplex (and would therefore probably not be effective on Broccoli). I do not know if dye labeling of oligonucleotides stabilizes them as well. The authors should at least discuss these issues because work using other systems (such as the MS2-GFP system, as documented recently by Parker and coworkers) can lead to unnaturally stable RNAs. This would be a concern if one is considering the use of the targeting aptamers to deliver RNAs that are not as stable as a G-quadruplex.

2. Although not central to the utility of the technology, the 1.2-fold enhancement of brightness of the 3WJdB module upon annealing with the targeting aptamer is curious. Is this difference (a) statistically significant? (b) does the effect persist if the broccoli domain is first annealed on its own, perhaps through more than one heat/cool cycle (perhaps with no Mg²⁺ at first, and then adding Mg²⁺) and then annealed to the targeting oligo?

3. Would the efficiency of internalization be higher if no excess of the targeting aptamer were present (it presumably competes with the assembled form)? Can the annealed complex be separated from the excess aptamer prior to application to cells?

Reviewer #2 (Remarks to the Author):

In this manuscript, Porciani et al. describe a platform for the targeted delivery of large RNA aptamer payloads specific cancer cell lines. The authors hybridize a cell targeting aptamer to a fluorogenic RNA aptamer through a linker "tail" region. The authors target the delivery of the RNA nanostructures to B cell leukemia cells and demonstrate the internalization and persistent function of folded RNAs greater than 200 nucleotides. The authors address the ability of large and small RNAs to be delivered specifically to the cells of interest, and maintain function within cells.

Overall, I believe this paper is a good fit for publication in Nature Communications after the authors address a few concerns. The impact of this paper is suitable for Nature Communications because it serves as an important stepping-stone for delivering nucleic acid aptamers from the extracellular solution to cytoplasm of cells of interest. This paper makes a case that aptamers can maintain their function through cell delivery and endocytosis.

The work has a few concerns that should be addressed before publication. Based on these issues, the manuscript needs to be revised before publication. My specific concerns are enumerated below.

1. One major claim is that large RNA molecules can be delivered to and function inside of cells using this platform. It is important for RNA chemical biologists interested in cellular delivery to understand the range of large RNA molecules that can be adopted for use with this system. To support the argument that large RNA aptamers can function in this platform, the authors should include experiments using additional payload aptamers with distinct structures or lengths.

2. It is important to characterize where the RNA molecules are localized after internalization. The authors address a related question regarding whether the targeting aptamers and payload aptamers colocalize. However, this question may not be as relevant to the potential users of the platform. Additionally, there is not sufficient explanation to resolve the discrepancies between correlation coefficients of the 3 different experimental conditions in figure 4. I am not convinced that brightness difference between AF488 and 3WJdB are able to explain the difference between panel b and c in figure 4. The authors should rule out some cleavage or other damage to RNA nanostructures.

3. The authors claim that delivery of aptamer-mediated RNA delivery is not affected by the size of the payload would be strengthened by the addition of an additional control. 3WJdB plus a control targeting aptamer should be compared to 3WJdB plus Waz or C10.36. AF488 anti-tail + control aptamer is compared with AF488 anti-tail + Waz or C10.36. However, the analogous control is not included for 3WJdB. 3WJdB+Waz and 3WJdB+C10.36 is instead compared to 3WJdB alone. Additional comparison in the presence of a control targeting aptamer would strengthen the conclusions drawn from figure 3 as well.

4. The authors do not adequately investigate the stability of RNA nanostructures over time, because they do not sample a long enough time frame, and there are not enough measurements in the early phase after washout. It is essential for RNA aptamer developers to understand how long delivered RNA aptamers can function within cells. The authors note a rapid reduction in 3WJdB signal as well as a gradual delay. The authors should work to further characterize both of these events. Additionally, as the authors mention in discussion, it would be especially important to further characterize the escape from endosomes.

Minor points:

1. The schematic representation of RNA nanostructures in figure 1 and other figures should clearly mark the 3' and 5' ends of RNA sequences.

2. I am somewhat confused by the three-way junction-containing Broccoli used here and the previous one described by Filonov et al., *Chemistry & Biology*, 2015. This latter paper developed a three-way junction, F30, and installed multiple Broccoli aptamers on each arm. The present one sounds very similar. Perhaps some clarification would be helpful - I assume the authors are using some type of improved variant of this earlier system.

Reviewer #3 (Remarks to the Author):

In this proof-of-concept study, the authors describe a modular aptamer nanostructure for aptamer-mediated delivery of larger functional RNA payloads (150-250 nt). The authors take advantage of previously characterized aptamers (transferrin receptor-targeting RNA aptamer and an aptamer directed against human cancer B cell lines) and functional RNA payloads (derivatives of the previously characterized fluorogenic broccoli aptamer). The authors show that different targeting and functional aptamers can be substituted thereby demonstrating the modularity of their nanostructure system. While this is an important aspect of the study and one that the authors highlight as a key advance, the concept of a modular aptamer delivery system is not novel

and was previously described by several groups including Eli Gilboa and John Rossi. In this later study, the concept of a 'sticky bridge' was presented for appending any small functional RNA to a targeting aptamer for targeted cell delivery. The authors go on to show that the aptamer nanostructure is taken up by target cells (transferrin receptor-positive cells or cancer B cell lines), likely by receptor mediated endocytosis. Importantly, once inside the cell (presumably endocytic compartment) the aptamer nanostructure retains its function as demonstrated by fluorescence of the spinach aptamer. While the potential implications of this study for delivery of larger RNAs (e.g. mRNAs – which are on average 10-fold longer than the functional RNA evaluated in this study) is significant, several points are missing or unclear.

1. The authors claim that the aptamer nanostructure system can be used to study the uptake and subcellular trafficking of functional RNAs, endosomal uptake and escape and cytoplasmic delivery. They claim that the subcellular localization of the aptamer nanostructure is within endosomes. While this is likely to be true based on previous aptamer delivery studies, the authors have failed to confirm that the aptamer nanostructure is indeed undergoing receptor-mediated uptake and that once inside the cells, it resides in endosomes. To confirm endosomal uptake, the authors should compete the RNA nanostructure using a ligand for the transferrin receptor (e.g. transferrin) or inhibitors. Endosomal targeting should be confirmed by performing co-staining with endocytic markers.
2. The use of cancer B cell lines, which are notorious for having large nuclei relative to cytoplasmic area, make it harder to determine the subcellular localization of the aptamer nanostructure. The authors claim the staining shows the aptamer nanostructure is perinuclear and consistent with endocytic retention, but this cannot be determined based on the cells that were used and the images presented. The authors should perform the subcellular localization studies with transferrin positive cells lines that have a more balanced nuclear to cytoplasmic area ratio to more easily resolve the subcellular location of the aptamer nanostructure.
3. Presumably, for delivery of large functional RNAs, the RNA cargo will have to be delivered to the cytosol. What percent of the input aptamer nanostructure makes it to the cytosol? Also, is the cargo (broccoli aptamer) still complexed with the targeting aptamer once in the cytosol? Previous studies by Dr. Kortylewski using CpG aptamers to deliver small functional RNAs (siRNAs) have demonstrated that the aptamer interacts with endosomal proteins. This interaction facilitates the release of the siRNA into the cytoplasm where it can be processed by the RNAi machinery.
4. The data in Figure 2 d and e require statistics
5. In the introduction, the authors suggest that their approach could be useful for delivery of large functional RNAs (mRNAs). Given the interest in mRNA delivery, the study would be strengthened if the authors could demonstrate delivery of a functional mRNA (e.g. GFP mRNA) to target cells.
6. The authors fail to reference the key proof-of-concept studies for the aptamer-mediated delivery of small functional RNAs (McNamara et al.) and aptamer delivery modular structures (Gilboa and Rossi).

Reviewer 1

The manuscript by Porciani et al. describes the application of aptamer nucleic acids that target specific cell surface receptors to deliver large non-coding RNAs to mammalian tissue culture cells. Previously, other groups have employed similar approaches to deliver small oligonucleotides, but this is, to my knowledge, the first application of this technology for delivering larger (> 100 nt) RNAs. The authors use dimeric and trimeric forms of the fluorogenic RNA broccoli as a cargo, and take advantage of the fluorescent properties of broccoli bound to its chromophore to report on cell internalization. The work appears technically sound in general, and I think it will be of interest to those in the RNA delivery and RNA therapeutics fields. I think the manuscript is topical and generally appropriate for this journal, but I have a couple of concerns that the authors may wish to address.

We thank the reviewer for his/her appreciation of our work.

1. As the authors point out, Broccoli and its relatives are structured around a G-quadruplex motif. These are known to be generally very stable secondary structure motifs, and to my knowledge all characterized cellular helicases that destabilize G-quadruplexes require a 3' single-stranded extension immediately adjacent to the G-quadruplex (and would therefore probably not be effective on Broccoli). I do not know if dye labeling of oligonucleotides stabilizes them as well. The authors should at least discuss these issues because work using other systems (such as the MS2-GFP system, as documented recently by Parker and coworkers) can lead to unnaturally stable RNAs. This would be a concern if one is considering the use of the targeting aptamers to deliver RNAs that are not as stable as a G-quadruplex.

We thank the reviewer for this observation. We included the following sentence in the fourth paragraph of Discussion that we believe can address the point raising by the reviewer:

“The longevity of RNAs inside the cells can vary depending on their structures or association with proteins to form RNP complexes⁶¹”

The new reference 61 corresponds to Garcia and Parker's paper (10.1261/rna.051797.115)] that the reviewer mentioned in his/her comment.

Moreover, to the best of our knowledge, dye labeling oligonucleotides should not reduce or prevent degradation of RNA mediated by exonucleases. However, to rule out any chance of increased resistance to exonucleases, our pulse-chase analysis (Fig. 6) was performed using a non-labeled 3WJdB (fully natural RNA). Therefore, both of its ends were unmodified and susceptible to enzymatic cleavage.

We also thank the reviewer for mentioning essential structural requirements needed by cellular helicase to unwind G-quadruplex sequences. 3WJdB bears indeed a 3' single-stranded extension, but is exploited to form a double stranded region with the targeting aptamer. This can potentially make 3WJdB less susceptible to cellular helicase. However, our data show that 3WJdB is located primarily in endosomes (see also new colocalization analyses in Figs. 4, 5, S14, and S15). To the best of our knowledge, only under particular conditions, helicases can be found in endosomes. One single report [Zhang et al. (10.1038/ni.2091)] described translocation of a specific helicase (DDX41) from the endoplasmic reticulum to endosomes of HEK293T cells after activation of Toll-like receptors by continuous stimulation (>4 hours) with poly(dA:dT). It is not known, however, whether this translocation is a cell line dependent mechanism, as other reports did not find helicases in endosomes. Because 3WJdB (but potentially any RNA payload of our nanostructure) is delivered to endosomes, its helicase-mediated destabilization in these vesicles should not occur during the timescale used in our pulse/chase analysis (1h pulse + 2h chase). In fact, even in the case of activation of Toll-like receptors (immune activation by RNAs can be advantageous for treating cancers), several hours (>4 hours) are required before DDX41

can translocate to endosomes. Therefore, **lysosomal nucleases are the main source of general degradation of RNAs in endocytic vesicles**. The long intracellular persistence of 3WJdB suggests that it avoids this degradation. Consistent with this interpretation, colocalization with Rab5 much more than with Rab7 suggests that most of the 3WJdB avoids the lysosomal degradation pathway by trafficking from one vesicle to another during the endosome maturation. This delays the RNA degradation that occurs in lysosomes (see also fourth paragraph of Discussion). These findings are in agreement with previous reports that have shown long stability of natural duplex siRNA (non-G quadruplex structure) located primarily in non-degradative vesicles (such as early and late endosomes but not lysosomes). Therefore, we believe that the long persistence of 3WJdB in NALM6 cells is not a peculiar feature of G-quadruplex-containing sequences.

2. Although not central to the utility of the technology, the 1.2-fold enhancement of brightness of the 3WJdB module upon annealing with the targeting aptamer is curious. Is this difference (a) statistically significant? (b) does the effect persist if the broccoli domain is first annealed on its own, perhaps through more than one heat/cool cycle (perhaps with no Mg²⁺ at first, and then adding Mg²⁺) and then annealed to the targeting oligo?

We thank the reviewer for this comment. We performed statistical analysis on the data shown in Fig 1d and reported statistical difference in a revised bar graph displayed in Fig. 1d.

A similar fluorescence enhancement between free 3WJdB and assembled 3WJdB was observed also upon annealing of 3WJdB with a 21-nt tail sequence (the same sequence presents at the 3'-end of Waz/C10.36). Therefore, we think that the extended sequence (anti-tail) at the 3'-end of 3WJdB may, to some extent, affect the overall RNA folding. However, upon hybridization with its complementary sequence, a more stable double-stranded region is generated, leading to enhanced folding stability and consequently to a slight (but statistically significant) fluorescence enhancement.

Similarly, in our previous report (see Alam et al. doi:10.1021/acssynbio.7b00059) we noted that the dimeric Broccoli aptamer (dB) with no additional RNA scaffold, such as tRNA or 3WJ, possessed an approximately 25% of signal relative to its monomeric version (mBroccoli). To increase folding stability of dB, we extended the terminal stem by adding four additional base pairings. Thanks to the presence of a more stable double-stranded domain, we generated a sequence, called "Stabilized dimeric Broccoli" (SdB), that showed a 2-fold enhanced fluorescence than mBroccoli.

3. Would the efficiency of internalization be higher if no excess of the targeting aptamer were present (it presumably competes with the assembled form)? Can the annealed complex be separated from the excess aptamer prior to application to cells?

We thank the reviewer for raising this question. The excess of free targeting aptamer can indeed compete with the assembled form in binding the target cell-surface receptor. In our manuscript, to maximize formation of aptamer complexes and reduce the fraction of free RNA payload, the annealing protocol was optimized using a 3:1 molar ratio of targeting:payload. However, further studies will be performed to fine-tune this molar ratio and find the best compromise between complex formation and binding of target cells.

Reviewer 2

In this manuscript, Porciani et al. describe a platform for the targeted delivery of large RNA aptamer payloads specific cancer cell lines. The authors hybridize a cell targeting aptamer to a

fluorogenic RNA aptamer through a linker “tail” region. The authors target the delivery of the RNA nanostructures to B cell leukemia cells and demonstrate the internalization and persistent function of folded RNAs greater than 200 nucleotides. The authors address the ability of large and small RNAs to be delivered specifically to the cells of interest, and maintain function within cells.

Overall, I believe this paper is a good fit for publication in Nature Communications after the authors address a few concerns. The impact of this paper is suitable for Nature Communications because it serves as an important stepping-stone for delivering nucleic acid aptamers from the extracellular solution to cytoplasm of cells of interest. This paper makes a case that aptamers can maintain their function through cell delivery and endocytosis.

The work has a few concerns that should be addressed before publication. Based on these issues, the manuscript needs to be revised before publication. My specific concerns are enumerated below.

We thank the reviewer for his/her overall appreciation of our work.

1. One major claim is that large RNA molecules can be delivered to and function inside of cells using this platform. It is important for RNA chemical biologists interested in cellular delivery to understand the range of large RNA molecules that can be adopted for use with this system. To support the argument that large RNA aptamers can function in this platform, the authors should include experiments using additional payload aptamers with distinct structures or lengths.

We thank the reviewer for these observations and suggestions. We believe that the manuscript has significantly advanced the field of cell targeting aptamers by demonstrating aptamer-mediated, targeted delivery of much larger RNAs than have been examined previously, with retention of RNA folding in the cytoplasm of target cells, and is thus already a significant contribution. The two-large payload RNAs studied here (176 and 244 nt) are, respectively, ~8- and ~12-fold larger than the size of an siRNA. We agree with the reviewer that the upper size limit for delivery with retention of biological activity has not been addressed. Probing those limits will be a major focus of future work.

2. It is important to characterize where the RNA molecules are localized after internalization. The authors address a related question regarding whether the targeting aptamers and payload aptamers colocalize. However, this question may not be as relevant to the potential users of the platform. Additionally, there is not sufficient explanation to resolve the discrepancies between correlation coefficients of the 3 different experimental conditions in figure 4. I am not convinced that brightness difference between AF488 and 3WJdB are able to explain the difference between panel b and c in figure 4. The authors should rule out some cleavage or other damage to RNA nanostructures.

We thank the reviewer for this suggestion. In response, new colocalization studies were performed using three different cancer cell lines (NALM6, HeLa, and MDA-MB-231) to assess sub-cellular localization of 3WJdB with three endocytic markers (Rab5, Rab7 and transferrin). As shown in new Figs. 4d, 4e, 4f, 5, S14, and S15, after 1h incubation our data indicate primary localization of this RNA payload in early endosomes and to a lesser extent in maturing endosomes. Thanks to these colocalization studies, we believe that the manuscript can offer a better understanding on the intracellular fate of the RNA payload. We also hope that our findings can offer useful hints to unveil further biological mechanisms that regulate trafficking of RNAs in endocytic vesicles and that can ultimately lead to long-lived RNAs (as reported for some natural siRNAs in previous works).

We also thank the reviewer for pointing out about the difference in Pearson's correlation coefficients calculated in Figs. 4a, 4b, and 4c. An improved colocalization between the two aptamer modules (targeting and payload) was indeed measured when a dye-labeled version of 3WJdB was used instead of monitoring fluorescence of 3WJd-DFHBI-1T complex. In the latter case, a high noise due to interactions of DFHBI-1T with intracellular components reduced the signal-to-noise ratio. In the first paragraph of the Results section entitled "Aptamer nanostructure is stably assembled upon endocytosis and localizes in endosomes", we included a new sentence emphasizing how the presence of high noise background can hamper a precise Pearson's correlation analysis leading to a value of correlation coefficient closer to 0 than it should be because the elevated background contributes to non-colocalized pixels (reference 38 was included to support it).

3. The authors claim that delivery of aptamer-medicated RNA delivery is not affected by the size of the payload would be strengthened by the addition of an additional control. 3WJdB plus a control targeting aptamer should be compared to 3WJdB plus Waz or C10.36. AF488 anti-tail + control aptamer is compared with AF488 anti-tail + Waz of C10.36. However, the analogous control is not included for 3WJdB. 3WJdB+Waz and 3WJdB+C10.36 is instead compared to 3WJdB alone. Additional comparison in the presence of a control targeting aptamer would strengthen the conclusions drawn from figure 3 as well.

We agree with the reviewer that it is important to utilize alternative delivery aptamers as controls. Fortunately, Waz and C10.36 can be used as controls for each other simply by changing the cells that are being targeted. For example, MOTN1 and HeLa express hTfR but not the target of C10.36, and 3WJdB was delivered to these cells by Waz but not by C10.36. Mouse cell line SP1 expresses neither surface ligand, and 3WJdB was not delivered to those cells by either C10.36 or Waz. For each combination of targeting aptamer and non-targeted cell type, our data indicate that there is no difference in cell binding between 3WJdB alone and 3WJdB annealed to control aptamer sequences.

To further support these findings, we performed new flow cytometry analysis to compare cell staining of MDA-MB-231 cell line after 1h incubation with either Waz-3WJdB-Cy3, ctrl Apt-3WJdB-Cy3, or free 3WJdB-Cy3. The presence of a non-targeting sequence annealed to 3WJdB does not increase the extent of non-specific binding compared to 3WJdB alone (making 3WJdB alone the proper control for our experiments). We included these new data in Figure S15a.

4. The authors do not adequately investigate the stability of RNA nanostructures over time, because they do not sample a long enough time frame, and there are not enough measurements in the early phase after washout. It is essential for RNA aptamer developers to understand how long delivered RNA aptamers can function within cells. The authors note a rapid reduction in 3WJdB signal as well as a gradual delay. The authors should work to further characterize both of these events. Additionally, as the authors mention in discussion, it would be especially important to further characterize the escape from endosomes.

We agree with the reviewer that there could be additional insights gained by either a fine-grained analysis of the initial time points (to calculate rate constants relative to initial drop and subsequent gradual reduction of fluorescence) or by including a longer chase phase. We chose not to pursue those questions for several reasons. First, the main goal of the pulse/chase analysis was to establish the overall trend of RNA persistence inside the cells upon internalization. This point is established by the data in Fig 6. Second, it seems likely that any rate constants that we measure for this system could differ significantly upon targeting a different surface receptor or upon delivering a payload that has more consequences for cell

biology than 3WJdB. Third, in presence of serum-containing medium, we noticed a certain increase of fluorescence in NALM6 cells treated with only DFHBI-1T at longer time points of our chase step (90 and 120 min). If the pulse-chase analysis is extended for additional hours, this variation of non-specific background can lead to misleading calculation of the real fraction of folded and fully-functional 3WJdB. We did not notice significant fluorescence changes in DFHBI-1T only-treated cells when incubated in a serum-free medium (Fig 6). However, after a total of 3 hrs (1hr pulse + 2hr chase) we ended the persistence studies to minimize changes in overall cellular responses due to the use of serum-free medium. Extending the time in serum-free medium would have increasingly become a study of cells under stress.

Minor points:

1. The schematic representation of RNA nanostructures in figure 1 and other figures should clearly mark the 3' and 5' ends of RNA sequences.

Corrected.

2. I am somewhat confused by the three-way junction-containing Broccoli used here and the previous one described by Filonov et al., *Chemistry & Biology*, 2015. This latter paper developed a three-way junction, F30, and installed multiple Broccoli aptamers on each arm. The present one sounds very similar. Perhaps some clarification would be helpful - I assume the authors are using some type of improved variant of this earlier system.

We independently designed and generated 3WJdB in our previous work (see ref 30: Alam et al. doi:10.1021/acssynbio.7b00059). To acknowledge the difference between our design and the one showed by Filonov et al. we included the following sentences in the first paragraph of Results that we believe can address the point raising by the reviewer:

“Filonov et al.⁴⁶ previously reported “F30–2xBroccli” that contains two Broccoli aptamers incorporated in arms 1 and 2 of the same 3WJ RNA scaffold. However, in our design, the two monomers are incorporated in arms 1 and 3, which are spatially oriented at approximately 180° from each other to reduce chromophore-chromophore and inter-helical RNA interactions between the two Broccoli monomers³⁰”.

Reviewer 3

In this proof-of-concept study, the authors describe a modular aptamer nanostructure for aptamer-mediated delivery of larger functional RNA payloads (150-250 nt). The authors take advantage of previously characterized aptamers (transferrin receptor-targeting RNA aptamer and an aptamer directed against human cancer B cell lines) and functional RNA payloads (derivatives of the previously characterized fluorogenic broccoli aptamer). The authors show that different targeting and functional aptamers can be substituted thereby demonstrating the modularity of their nanostructure system. While this is an important aspect of the study and one that the authors highlight as a key advance, the concept of a modular aptamer delivery system is not novel and was previously described by several groups including Eli Gilboa and John Rossi. In this later study, the concept of a ‘sticky bridge’ was presented for appending any small functional RNA to a targeting aptamer for targeted cell delivery.

The authors go on to show that the aptamer nanostructure is taken up by target cells (transferrin receptor-positive cells or cancer B cell lines), likely by receptor mediated endocytosis. Importantly, once inside the cell (presumably endocytic compartment) the aptamer nanostructure retains its function as demonstrated by fluorescence of the spinach aptamer.

While the potential implications of this study for delivery of larger RNAs (e.g. mRNAs – which are on average 10-fold longer than the functional RNA evaluated in this study) is significant, several points are missing or unclear.

We thank the reviewer for his/her overall appreciation of our work.

1. The authors claim that the aptamer nanostructure system can be used to study the uptake and subcellular trafficking of functional RNAs, endosomal uptake and escape and cytoplasmic delivery. They claim that the subcellular localization of the aptamer nanostructure is within endosomes. While this is likely to be true based on previous aptamer delivery studies, the authors have failed to confirm that the aptamer nanostructure is indeed undergoing receptor-mediated uptake and that once inside the cells, it resides in endosomes. To confirm endosomal uptake, the authors should compete the RNA nanostructure using a ligand for the transferrin receptor (e.g. transferrin) or inhibitors. Endosomal targeting should be confirmed by performing co-staining with endocytic markers.
2. The use of cancer B cell lines, which are notorious for having large nuclei relative to cytoplasmic area, make it harder to determine the subcellular localization of the aptamer nanostructure. The authors claim the staining shows the aptamer nanostructure is perinuclear and consistent with endocytic retention, but this cannot be determined based on the cells that were used and the images presented. The authors should perform the subcellular localization studies with transferrin positive cells lines that have a more balanced nuclear to cytoplasmic area ratio to more easily resolve the subcellular location of the aptamer nanostructure.

These two points both address intracellular localization. We thank the reviewer for the important suggestion in point #2. In response to this comment, new colocalization studies were performed to assess sub-cellular localization of 3WJdB in the original B cell leukemia cell line (NALM6), and in two transferrin receptor-positive cell lines (HeLa and MDA-MB-231) that have a more balanced ratio of nuclear-to-cytoplasmic area. For each cell line, co-localization was evaluated with respect to three endocytic markers (Rab5, Rab7 and transferrin). As shown in Figs. 4d, 4e, 4f, 5, S14, and S15, after 1h incubation 3WJdB was localized preferentially at the perinuclear region of these cells in Rab5- and Tf-containing vesicles in early endosomes and to a lesser extent in Rab7-containing vesicles in maturing endosomes. In combination with our original data (such as blocking of observed cellular uptake when cells were incubated on ice and correlation between Waz-dependent cell staining and TfR expression as measured by anti-CD71 Ab), the strong intracellular colocalization with transferrin and Rab5 further establish that Waz-3WJdB uptake is indeed due to TfR-mediated internalization via receptor-mediated endocytosis and that endosome trafficking and maturation strongly contributes to the long persistence of this RNA payload observed in our pulse-chase analysis (Fig. 6).

3. Presumably, for delivery of large functional RNAs, the RNA cargo will have to be delivered to the cytosol. What percent of the input aptamer nanostructure makes it to the cytosol? Also, is the cargo (broccoli aptamer) still complexed with the targeting aptamer once in the cytosol? Previous studies by Dr. Kortylewski using CpG aptamers to deliver small functional RNAs (siRNAs) have demonstrated that the aptamer interacts with endosomal proteins. This interaction facilitates the release of the siRNA into the cytoplasm where it can be processed by the RNAi machinery.

We agree with the reviewer that there are multiple applications for which accessing the cytosol is a prerequisite for biological activity. Indeed, improving cytosolic escape is perhaps the single greatest barrier to the field of oligonucleotide therapeutics. (It would be a tall order to expect this study to resolve that larger question!) Instead, the main goal of this work was to determine whether the cell-internalizing properties of aptamers that have been exploited to deliver small RNAs could be extended to larger RNA payloads. That effort was successful. In addition, our

observations of internalization into endosomes and persistence for several hours within those vesicles have two important implications. The first is that large RNAs delivered into endosomes may have a sizable time window during which to escape into the cytosol. Nanostructure components designed to aid that translocation may therefore be able to do so over an extended period of time. The second is that endosomes are, themselves, a targetable biological compartment. Future work will address whether RNA aptamers with biological activity are able to interfere and alter the biology of target cells by acting either at the cytosolic or endosomal level.

4. The data in Figure 2 d and e require statistics

We thank the reviewer for his/her comment. We performed statistical analysis of our flow cytometry data and relative results were included in Fig 2d and e.

5. In the introduction, the authors suggest that their approach could be useful for delivery of large functional RNAs (mRNAs). Given the interest in mRNA delivery, the study would be strengthened if the authors could demonstrate delivery of a functional mRNA (e.g. GFP mRNA) to target cells.

We agree that such a demonstration would be a major finding. As mentioned in the answer to point 3 above, endosomal escape of large RNA payloads is a separate question that is beyond the scope of the present work. We also agree that future studies should be performed to assess and improve cytosolic accessibility of therapeutic RNAs (such as mRNAs or RNA aptamers) and to alter the biology of target cells.

6. The authors fail to reference the key proof-of-concept studies for the aptamer-mediated delivery of small functional RNAs (McNamara et al.) and aptamer delivery modular structures (Gilboa and Rossi).

We thank the reviewer for this observation. We were aware of these papers and included them in early drafts. It was an oversight not to have noticed that they were deleted during the editing phase of manuscript preparation. We have restored those references (see references 41, 42, 43) in the revised version.

REVIEWERS' COMMENTS:

Reviewer #1 (Remarks to the Author):

The authors have addressed all my concerns.

Reviewer #2 (Remarks to the Author):

I think most of the key points have been addressed, and the manuscript is improved. I don't have additional comments.

Reviewer #3 (Remarks to the Author):

While the authors have adequately addressed many of the reviewer's concerns, one of the remaining issues is demonstration, at least in vitro, that their system is capable of delivering longer RNAs (the sizes of mRNAs). This issue was raised by several reviewers (myself and also reviewer 2).

While in their rebuttal the authors claim that the major objective of their work was to show that aptamers can be used to deliver longer RNAs, in the manuscript the major objective is stated as delivery of long therapeutic RNAs (e.g. mRNAs). While the sizes of the RNAs that are successfully delivered are ~200nt, the authors still claim, in the intro, that the technology could be used to deliver larger therapeutic RNAs (mRNAs) as this increases the impact of their work.

The authors also fail to give due credit for the concept of modular aptamers delivery payloads. As I previously noted, while this is an important aspect of the study and one that the authors highlight as a key advance, the concept of a modular aptamer delivery system is not novel and was previously described by several groups including Eli Gilboa and John Rossi. In this later study, the concept of a 'sticky bridge' was presented for appending any small functional RNA to a targeting aptamer for targeted cell delivery.

Reviewer 3

While the authors have adequately addressed many of the reviewer's concerns, one of the remaining issues is demonstration, at least in vitro, that their system is capable of delivering longer RNAs (the sizes of mRNAs). This issue was raised by several reviewers (myself and also reviewer 2).

While in their rebuttal the authors claim that the major objective of their work was to show that aptamers can be used to deliver longer RNAs, in the manuscript the major objective is stated as delivery of long therapeutic RNAs (e.g. mRNAs). While the sizes of the RNAs that are successfully delivered are ~200nt, the authors still claim, in the intro, that the technology could be used to deliver larger therapeutic RNAs (mRNAs) as this increases the impact of their work.

Our system might or might not deliver mRNAs or other similarly-sized RNAs, and we never intended to imply that we have demonstrated such delivery. We searched the document to text that might have suggested such indirect claims, and adjusted these to tone them down, both in the introduction and in the discussion.

Here are the revised sentences.

INTRODUCTION SECTION

Original sentence in the second paragraph:

"However, with the advent of the CRISPR/cas9 revolution and the growing interest in aptamers and mRNAs to modulate biological processes,...."

Revised sentence:

"However, with the advent of the CRISPR/cas9 revolution and the growing interest in aptamers and other RNAs to modulate biological processes,...."

Original sentence in the third paragraph:

"We show here that fluorogenic RNA aptamers can be used as surrogates for other large RNA payloads to accelerate screening of nanostructure designs"

Revised sentence:

We show here that fluorogenic RNA aptamers can be used as surrogates for other large RNA payloads with comparable size to accelerate screening of nanostructure designs

Original sentence in the last paragraph:

"This work highlights the application of fluorogenic RNA aptamers as real-time reporters to assess retention of the correct aptamer folding within the nanostructure both after assembly and upon endocytosis into B cell leukemia cell lines, thus verifying the effective aptamer-mediated targeted delivery of large functional RNAs."

Revised sentence:

"...thus verifying that aptamers can mediate effective, targeted delivery of much larger functional RNAs than has previously been reported."

DISCUSSION SECTION

Original sentence in the first paragraph:

"The fluorogenic properties of the enhanced Broccoli aptamer variants were used with a dual function: (i) to assess retention and persistence of aptamer payload folding when assembled into a more complex RNA structure both in vitro and in live cells, and (ii) to demonstrate aptamer-mediated targeted delivery of large functional RNAs in cancer cells."

Revised sentence:

"... and (ii) to demonstrate aptamer-mediated targeted delivery of large functional RNAs (at least 244nt; 80kDa) in cancer cells."

The authors also fail to give due credit for the concept of modular aptamers delivery payloads. As I previously noted, while this is an important aspect of the study and one that the authors highlight as a key advance, the concept of a modular aptamer delivery system is not novel and was previously described by several groups including Eli Gilboa and John Rossi. In this later study, the concept of a 'sticky bridge' was presented for appending any small functional RNA to a targeting aptamer for targeted cell delivery.

The reviewer may have missed the statement below that cites Gilboa and Rossi's papers:

This nanostructure displays a targeting aptamer module and a payload aptamer module, and the two modules self-assemble via a double-stranded connector sequence similarly to previous aptamer-siRNA or bispecific aptamer hybrids^{13,40-42}